# WHEN TO RETRAIN AFTER DRIFT: A DATA-ONLY TEST OF POST-DRIFT DATA SIZE SUFFICIENCY

**Ren Fujiwara[1], Yasuko Matsubara[1], Yasushi Sakurai[1],**
[1]SANKEN, The University of Osaka, Japan
`{r-fujiwr88,yasuko,yasushi}@sanken.osaka-u.ac.jp`

## ABSTRACT

Sudden concept drift makes previously trained predictors unreliable, yet deciding when to retrain and what post-drift data size is sufficient is rarely addressed. We propose CALIPER—a detector- and model-agnostic, data-only test that estimates the post-drift data size required for stable retraining. CALIPER exploits state dependence in streams generated by dynamical systems: we run a single-pass weighted local regression over the post-drift window and track a one-step proxy error as a function of a locality parameter $\theta$. When an effective sample size gate is satisfied, a monotonically non-increasing trend in this error with increasing a locality parameter indicates that the data size is sufficiently informative for retraining. We also provide a theoretical analysis of our method, and we show that the algorithm has a low per-update time and memory. Across datasets from four heterogeneous domains, three learner families, and two detectors, CALIPER consistently matches or exceeds the best fixed data size for retraining while incurring negligible overhead and often outperforming incremental updates. CALIPER closes the gap between drift detection and data-sufficient adaptation in streaming learning.

## 1 INTRODUCTION

Despite the ubiquity of data streams, building reliable time-series predictors in non-stationary environments remains challenging. A substantial body of work shows that maintaining performance hinges on rapid adaptation to concept drift (Gama et al., 2004; Baena-García et al., 2006; Bifet and Gavaldà, 2007; Frias-Blanco et al., 2014; Sebastião and Fernandes, 2017; Raab et al., 2020; Pham et al., 2023; Kawabata et al., 2023; Zhang et al., 2023; Higashiguchi et al., 2025; Chihara et al., 2025; Matsubara and Sakurai, 2025; Zhao and Shen, 2025; Verma et al., 2025). However, most practical gains are achieved under incremental drift, where the data distribution changes gradually. In contrast, real-world streams often undergo abrupt shifts that invalidate previously learned models (Hare and Mantua, 2000; Folke et al., 2004; Matsubara and Sakurai, 2016). When such a sudden drift occurs, a pragmatic and effective remedy is to retrain the predictor on newly arrived post-drift data rather than salvaging the pre-drift model (Gama et al., 2014; Lu et al., 2017). We focus on this sudden drift regime and study how to provide stable retraining—namely, how to determine the post-drift data size needed to restore accuracy safely.

Under such sudden drift, window-based strategies are widely used in streaming settings. ADaptive WINdowing (ADWIN) (Bifet and Gavaldà, 2007) overcomes the limitations of fixed windows by dynamically splitting a sliding window into two subwindows and provides provable bounds on false positives and false negatives, which is why it is widely adopted. Kolmogorov–Smirnov WINdowing (KSWIN) (Raab et al., 2020) applies a two-sample Kolmogorov–Smirnov test over empirical cumulative distribution functions (CDFs) to capture distributional changes beyond mean shifts, including variance and shape. However, detection alone does not tell us what post-drift data size is needed to retrain a model that will generalize. Updating too early risks overfitting to transient noise; waiting too long prolongs downtime, keeps a stale pre-drift model in production for an extended period, and degrades predictive accuracy. These trade-offs call for a principled way to decide when the post-drift data size has become sufficient to retrain safely.

We therefore deliberately focus on a different question than classical drift detection: given that a drift alarm has already been raised, how many post-drift samples are needed to safely retrain the

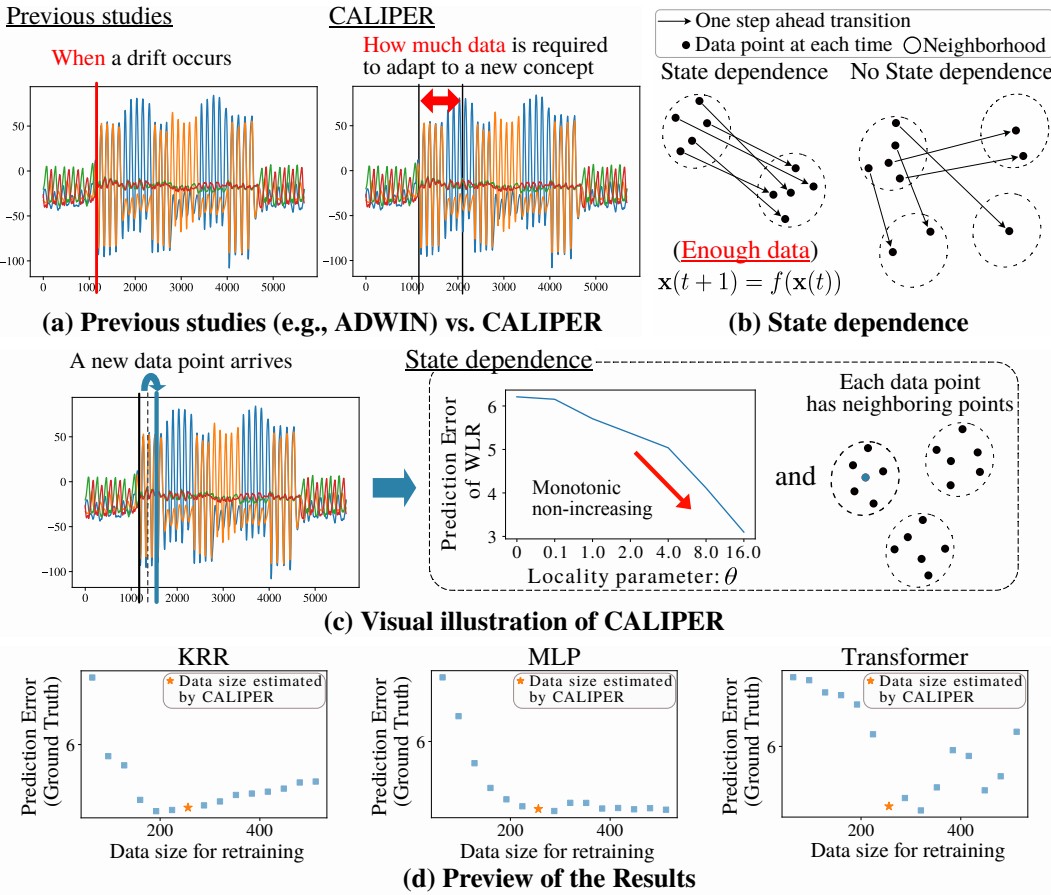

Figure 1: **Overview of CALIPER. (a) Unlike window-based detectors (e.g., ADWIN/KSWIN) that only indicate if/when drift occurs, CALIPER estimates how much post-drift data are needed for stable retraining. (b) State dependence: for dynamical systems $(\mathbf{x}_{t+1} = f(\mathbf{x}_t))$, nearby states exhibit similar one-step transitions; thus data sufficiency reduces to testing whether the post-drift window exhibits adequate state dependence. (c) Pipeline: a locality parameter $\theta$ reweights nearby samples in weighted local regression; when the proxy error is monotonically non-increasing as $\theta$ increases and the neighborhood is sufficiently populated, retraining is triggered. (d) Results: star markers denote CALIPER's estimated data sizes that yield low post-drift errors across heterogeneous learners (Kernel Ridge Regression (KRR), MLP, Transformer). CALIPER selects the optimal post-drift data size—i.e., the point at which retraining would be stable—without any retraining.**

downstream model? Window-based detectors such as ADWIN and KSWIN decide whether and when a drift occurs, but they are silent about the post-drift data sufficiency problem. In practice, this gap is critical: choosing a post-drift window that is too short leads to unstable retraining and oscillations, whereas waiting for an overly long window prolongs the use of a stale pre-drift model. Moreover, repeated probe-and-train approaches that retrain complex predictors (e.g., deep neural networks (DNNs)) to gauge readiness are computationally prohibitive in streaming scenarios. This leads to our central question: Can we estimate the post-drift data size requirement directly from the stream, without actually retraining the model?

To answer this, we propose CALIPER, *Cumulative Assessment of Locality Indicator for Post-drift Estimation of Retraining-size*. Assuming the data are generated by a (possibly nonlinear) dynamical system, CALIPER exploits state dependence to infer a sufficient post-drift data size without retraining. Concretely, it partitions the post-drift window into a reference set and test points, and tracks a self-supervised proxy prediction error (e.g., one-step-ahead error) from a weighted local regression whose weights are governed by a locality parameter $\theta$. We apply an exponentially decaying

kernel parameterized by $\theta$ to distances in feature space between test points and reference samples. A monotonically non-increasing trend in proxy error with increasing $\theta$ indicates adequate state dependence and triggers a retraining decision. The procedure is single-pass and computationally efficient, leveraging dynamical structure rather than model-specific internals. Finally, our analysis links CALIPER's trigger to a formal notion of state dependence: under a stylized dynamical model, passing the monotone-locality test implies stronger state dependence. We also provide an interpretation, via data-dependent generalization bounds, suggesting that stronger state dependence can be favorable for stable retraining.

**Overview of CALIPER.** Fig. 1 summarizes CALIPER: unlike window-based detectors such as ADWIN/KSWIN that merely signal whether/when a change occurs, CALIPER estimates how much post-drift data are sufficient for stable retraining. The key insight is state dependence: if the data stream follows a dynamical law $(\mathbf{x}(t+1) = f(\mathbf{x}(t)))$, nearby states exhibit similar one-step transitions, so deciding sufficiency reduces to testing whether the post-drift window displays adequate local consistency. Operationally, CALIPER probes this via a locality parameter $\theta$ that upweights nearby samples in a weighted local regression; a monotonically non-increasing proxy prediction error as $\theta$ increases—together with a sufficiently populated neighborhood—certifies a data-side sufficiency proxy (ESS + monotone locality curve) and triggers retraining. As previewed in panel (d), the star-marked data sizes selected by CALIPER yield few post-drift test errors across heterogeneous learners (Kernel Ridge Regression (KRR), MLP, Transformer). In contrast, excessively large data sizes worsen the error by delaying the update and prolonging the use of a stale model. Crucially, CALIPER produces these estimates without observing at the post-drift test segment or relying on model-specific internals, closing the gap between drift detection and data-sufficient adaptation.

Our key contributions can be summarized as follows:

- **Problem & Method.** We formalize post-drift data sufficiency—estimating the minimum window size needed to safely retrain after a sudden drift, given an external drift alarm. In contrast to classical drift detectors, which only decide whether and when a change occurred, our focus is on how much post-drift data are required for stable adaptation. We propose CALIPER, a detector- and model-agnostic, data-only procedure that selects the earliest window that passes the effective sample size (ESS) gate and monotone locality test over a single-pass weighted local regression.

- **Effective and Efficient.** We show that CALIPER can determine whether the data exhibits state dependence. We provide an interpretation, via data-dependent generalization bounds, suggesting that stronger state dependence can be favorable for stable retraining under standard regularity conditions. The algorithm is streaming-friendly: it runs in a single pass and keeps per-update time and memory costs low by solving small weighted regressions under a fixed locality schedule.

- **Empirical validation.** Across four datasets (MoCap, TEP, Automobile, Dysts), three model families (KRR, MLP, Transformer), and two detectors (ADWIN, KSWIN), CALIPER matches or exceeds the best fixed data size retraining without per-dataset tuning, improves post-drift error and recovery, and outperforms incremental updates with a negligible overhead.

## 2 PROPOSED METHOD: CUMULATIVE ASSESSMENT OF LOCALITY INDICATOR FOR POST-DRIFT ESTIMATION OF RETRAINING-SIZE (CALIPER)

### 2.1 PROBLEM DEFINITION

We consider a multivariate data stream $\{\mathbf{x}(t) \in \mathbb{R}^d\}_{t \geq 1}$ monitored by a drift detector. When a drift is detected at time $t_s$, we focus on the post-drift portion of the stream.

**Definition 1** (Post-drift window and data size). *For any $t \geq t_s+1$, the post-drift window is the set of observed samples*

$$\mathbf{X}_t = \{\mathbf{x}(t_s), \mathbf{x}(t_s+1), \ldots, \mathbf{x}(t)\},$$

*and its data size (window length) is the cardinality*

$$n_t = |\mathbf{X}_t| = t - t_s + 1.$$

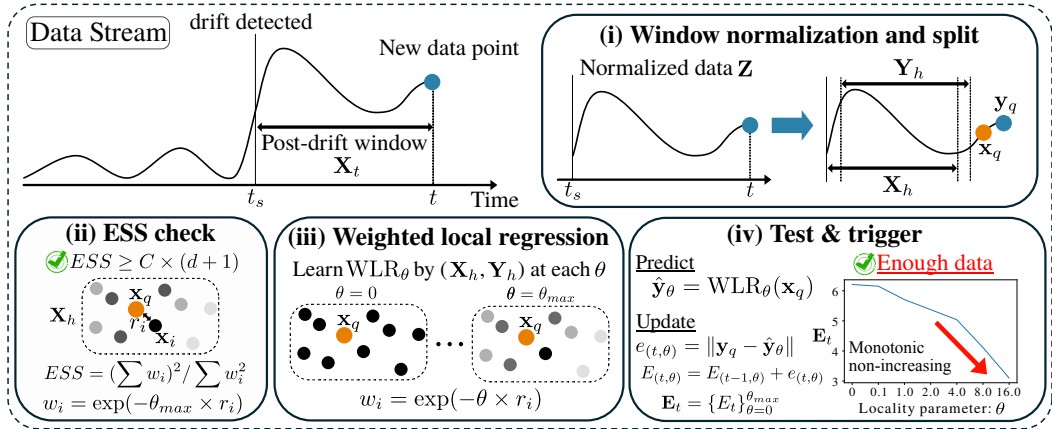

**Figure 2: CALIPER, a model-agnostic framework for dynamically estimating the data size required for retraining after sudden concept drift in a data stream: (i) Window normalization and split: after a drift alarm, the post-drift segment is normalized and partitioned into reference pairs $(\mathbf{X}_h, \mathbf{Y}_h)$ and a query $(\mathbf{x}_q, \mathbf{y}_q)$. (ii) ESS check: kernel weights $\mathbf{w}_\theta = \exp(-\theta \times \mathbf{r})$ at the largest $\theta$ define an effective neighborhood; proceed only if $ESS \geq C \times (d+1)$. (iii) Weighted local regression: for each $\theta$ on a fixed grid, solve the weighted normal equations to obtain $\hat{\mathbf{y}}_\theta$ and compute a proxy prediction error. (iv) Test and trigger: a monotonic non-increase of the error as $\theta$ increases, sustained for consecutive updates, indicates (a proxy for) sufficient local regularity/state dependence and triggers retraining. The rightmost panel illustrates the monotone trend of error versus $\theta$.**

The downstream predictor is treated as a black box—we do not access its internals. Intuitively, as data arrive, we wish to decide the minimum post-drift data size that allows a full stable retraining, avoiding triggers that are too early (overfitting/oscillation) or too late (prolonged degradation).

**Problem 1** (Post-Drift Data Sufficiency). *Given the post-drift stream after an alarm at $t_s$, find the smallest $n^\star \in \mathbb{N}$ such that retraining the downstream predictor on $\mathbf{X}_{t_s+n^\star}$ is stable, using only the observed post-drift window (no access to model internals or post-drift test labels). Ideally,*

$$n^\star = \min\{\, n \geq 1 : \Pr(\mathcal{L}_{\text{gen}}(f_{\mathbf{X}_{t_s+n}}) \leq \varepsilon) \geq 1 - \delta \,\},$$

*where $f_\mathbf{X}$ is the predictor retrained on $\mathbf{X}$, and $\mathcal{L}_{\text{gen}}$ denotes post-drift generalization loss. Because $\mathcal{L}_{\text{gen}}$ is unobservable online, the problem reduces to designing a model-agnostic, data-side stopping criterion $R(\mathbf{X}_t) \in \{0, 1\}$ and selecting*

$$n^\star = \min\{\, n \geq 1 : R(\mathbf{X}_{t_s+n}) = 1 \,\}.$$

This definition is intentionally idealized: in a streaming setting the post-drift generalization loss $L_{\text{gen}}$ is not observable online, and thus $n^\star$ cannot be computed directly. Instead, we must rely on a data-side surrogate that can be evaluated on the post-drift window alone. Concretely, our goal is to design a model-agnostic stopping rule $R(X_t) \in \{0, 1\}$ such that the smallest time $t$ with $R(X_t) = 1$ closely approximates the ideal $n^\star$ in Problem 1. The subsequent sections construct such a criterion based on state dependence and effective sample size and establish conditions under which it reliably predicts sufficient post-drift data for retraining.

The next subsection introduces our method: a concrete stopping criterion $R(\cdot)$ and an efficient online algorithm with which to estimate it from the stream and trigger stable retraining—both detector- and model-agnostic, and requiring no post-drift labels.

## 2.2 CALIPER: Cumulative Assessment of Locality Indicator for Post-drift Estimation of Retraining Data Size

We specify (i) a concrete data-side criterion $R(\cdot)$ for stable retraining and (ii) an efficient online algorithm that sequentially estimates $R(\cdot)$ from the stream, resolving the early/late trigger trade-off in a data-driven manner. In Appendix A, we provide Algorithm 1, which is an overview of our algorithm.

### 2.2.1 STOPPING CRITERION

We define a model-agnostic, data-side stopping criterion $R(\mathbf{X}_t) \in \{0, 1\}$ on the post-drift window $\mathbf{X}_t = \{\mathbf{x}(t_s), \ldots, \mathbf{x}(t)\}$. The criterion fires when (a) the current neighborhood is sufficiently populated and (b) increasing locality consistently improves one-step predictability, indicating state dependence and local regularity. When $R(\mathbf{X}_t) = 1$, we trigger stable retraining on $\mathbf{X}_t$.

### 2.2.2 ONLINE ESTIMATION ALGORITHM (CALIPER)

At each $t \geq t_s + 1$ we execute the following four steps. All operations in Steps (ii) and (iii) are performed for each value of the locality parameter $\theta$ in a fixed grid $\Theta = \{\theta_0, \ldots, \theta_{\max}\}$, yielding a sequence of localized predictors and prediction errors indexed by $\theta$.

**(i) Window Normalization and Split.** We normalize the post-drift window to obtain $Z \in \mathbb{R}^{n_t \times d}$. This normalization is applied within the current post-drift window and is used only to stabilize distances for the locality kernel; CALIPER never compares distances across different windows, so the underlying drift dynamics are not masked by this rescaling. Normalize the post-drift window to obtain $\mathbf{Z} \in \mathbb{R}^{n_t \times d}$. Define

$$\mathbf{X}_h = \mathbf{Z}[1{:}(n_t{-}2)], \quad \mathbf{Y}_h = \mathbf{Z}[2{:}(n_t{-}1)], \quad (\mathbf{x}_q, \mathbf{y}_q) = (\mathbf{z}(n_t{-}1), \mathbf{z}(n_t)).$$

Thus $(\mathbf{X}_h, \mathbf{Y}_h)$ provides $n_t{-}2$ reference pairs $(\mathbf{z}(s), \mathbf{z}(s{+}1))$, and $(\mathbf{x}_q, \mathbf{y}_q)$ is the current query pair.

**(ii) ESS Check.** Fix a short locality grid $\Theta = \{0, \ldots, \theta_{\max}\}$ with $\theta_{\max}$ giving the tightest locality. Let the raw distances be $r_i^{\mathrm{raw}} = \|\mathbf{X}_h^{(i)} - \mathbf{x}_q\|$ and define $D = \mathrm{mean}(\{r_i^{\mathrm{raw}}\}_i)$. Using the scaled distances $r_i = r_i^{\mathrm{raw}}/D$ (these $r_i$ are sample-to-query distances, distinct from the effective radius $r^{\mathrm{eff}}(\theta; \tau)$), set kernel weights $w_i(\theta) = \exp(-\theta\, r_i)$. Compute the effective sample size (ESS) at the tightest locality:

$$\mathrm{ESS}(\theta_{\max}) := \frac{\left(\sum_i w_i(\theta_{\max})\right)^2}{\sum_i w_i(\theta_{\max})^2}.$$

Proceed only if $\mathrm{ESS}(\theta_{\max}) \geq C\,(d{+}1)$. Because the kernel weights are $w_i(\theta) = \exp(-\theta r_i)$ with $r_i \geq 0$, the effective sample size $\mathrm{ESS}(\theta)$ is monotonically non-increasing in $\theta$: larger $\theta$ concentrates more weight on fewer neighbors. As a consequence, $\mathrm{ESS}(\theta_{\max})$ is the smallest ESS value on the grid. Checking the gate only at $\theta_{\max}$ therefore guarantees that $\mathrm{ESS}(\theta_k) \geq C(d + 1)$ for all $\theta_k \leq \theta_{\max}$.

**(iii) Weighted Local Regression (WRL).** We fit a lightweight weighted local regression model around the current query point, using kernel weights $w_i(\theta)$ to emphasize nearby samples. Augment references with a bias: $\mathbf{X}_{\mathrm{aug}} = [\mathbf{X}_h \mid \mathbf{1}] \in \mathbb{R}^{(n_t-2)\times p}$ with $p = d{+}1$, and let $\mathbf{x}_{\mathrm{aug}} = [\mathbf{x}_q \mid 1]$. For each $\theta \in \Theta$, form

$$\mathbf{W}_\theta = \mathrm{diag}(w_i(\theta)), \quad \mathbf{A}_\theta = \mathbf{X}_{\mathrm{aug}}^\top \mathbf{W}_\theta \mathbf{X}_{\mathrm{aug}}, \quad \mathbf{B}_\theta = \mathbf{X}_{\mathrm{aug}}^\top \mathbf{W}_\theta \mathbf{Y}_h,$$

solve the small system $\boldsymbol{\beta}_\theta = \mathbf{A}_\theta^{-1} \mathbf{B}_\theta$.

**(iv) Test & Trigger.** Compute the query prediction and one-step proxy error

$$\hat{\mathbf{y}}_\theta = \mathbf{x}_{\mathrm{aug}}^\top \boldsymbol{\beta}_\theta, \qquad e_{(t,\theta)} = \|\mathbf{y}_q - \hat{\mathbf{y}}_\theta\|.$$

Accumulate the proxy error in the original units,

$$E_{(t,\theta)} = E_{(t-1,\theta)} + e_{(t,\theta)}.$$

A smaller $\theta$ yields broader (more global) averaging, whereas a larger $\theta$ focuses on nearer neighbors; under state dependence, increasing $\theta$ should reduce error until neighborhoods become too sparse—an effect controlled by the ESS gate. Finally, on the ordered grid $\Theta = \{\theta_k\}$, test monotonicity:

$$E_{(t,\theta_k)} \geq E_{(t,\theta_{k+1})} \quad \forall k.$$

If the test holds, set $R(\mathbf{X}_t) = 1$ and trigger retraining on the current post-drift window $\mathbf{X}_t$; otherwise, continue streaming.

## 2.3 THEORETICAL ANALYSIS FOR CALIPER

We provide formal guarantees explaining why the CALIPER introduced in this work is useful for estimating the amount of data needed for retraining after sudden concept drift. In particular, we link CALIPER's trigger—monotonicity of localized one-step prediction error under a sufficient effective sample size (ESS)—to a rigorous notion of state dependence, and we provide an interpretation for why stronger state dependence can correlate with more stable retraining on an appropriate local region. We begin by formalizing the setting and introducing the key quantities used in our analysis.

**Setting.** We consider a $d$-dimensional time series $\{\mathbf{s}(t)\}_{t \geq 0} \subset \mathbb{R}^d$ generated by

$$\mathbf{s}(t+1) \; = \; f(\mathbf{s}(t)) + \xi_t, \qquad t = 0, 1, 2, \dots \tag{1}$$

where $f : \mathbb{R}^d \to \mathbb{R}^d$ is locally $L$-Lipschitz and $\{\xi_t\}$ is a zero-mean noise process with sub-Gaussian coordinates and covariance bounded by $\sigma^2 I_d$. When needed for concentration, we assume $\beta$-mixing with summable coefficients.

**Locality parameterization ($\theta$ vs. effective radius).** CALIPER is implemented using the locality parameter $\theta$ through the kernel $w_i(\theta) = \exp(-\theta \|\mathbf{s}_i - \mathbf{s}\|)$ (after window-wise normalization/scaling). For the theoretical analysis, it is sometimes convenient to index locality by an equivalent radius. Fix a threshold $\tau \in (0, 1)$ and define the effective radius

$$r^{\mathrm{eff}}(\theta; \tau) \; := \; \frac{\log(1/\tau)}{\theta} \qquad (\theta > 0),$$

so that $\exp(-\theta r) \leq \tau$ holds for all $r \geq r^{\mathrm{eff}}(\theta; \tau)$. Thus increasing $\theta$ corresponds to tighter locality (smaller effective radius). Accordingly, the implementation grid $\Theta = \{\theta_k\}_{k=1}^K$ induces a radius grid $\{r_k\}_{k=1}^K$ via $r_k = r^{\mathrm{eff}}(\theta_k; \tau)$ (with $\theta \to 0$ interpreted as the global limit). In what follows, we use $\theta_k$ and the corresponding $r_k$ interchangeably.

**Definition 2** (State dependence). *For $\mathbf{s} \in \mathbb{R}^d$ and radius $r > 0$, and for a fixed constant $c \geq L$, define*

$$\alpha(\mathbf{s}, r; c) \; := \; \Pr(\|\mathbf{s}'_+ - \mathbf{s}_+\| \leq c\, r \mid \|\mathbf{s}' - \mathbf{s}\| \leq r), \tag{2}$$

*where $\mathbf{s}'_+ = f(\mathbf{s}') + \xi'$ and $\mathbf{s}_+ = f(\mathbf{s}) + \xi$. Intuitively, a neighborhood that typically remains a neighborhood (up to factor $c$) after one step exhibits state dependence. Given a compact set $B \subset \mathbb{R}^d$, we say a window $\mathbf{S} = \{\mathbf{s}(t)\}_{t \in \mathcal{I}}$ exhibits state dependence on $B$ at scale $r$ if*

$$\inf_{\mathbf{s} \in B} \alpha(\mathbf{s}, r; c) \; \geq \; \underline{\alpha} \quad \text{for some } \underline{\alpha} \in (0, 1).$$

**Proposition 1** (CALIPER-triggered windows exhibit stronger state dependence). *Fix a compact set $B \subset \mathbb{R}^d$ and a constant $c \geq L$ as in equation 2. Let $\Theta = \{\theta_k\}_{k=1}^K$ be CALIPER's locality grid ordered as $0 < \theta_1 < \cdots < \theta_K = \theta_{\max}$, and let $\{r_k\}$ be the induced effective-radius grid defined above by $r_k = r^{\mathrm{eff}}(\theta_k; \tau)$ for a fixed $\tau \in (0, 1)$, so that $r_1 > \cdots > r_K = r_{\min}$. Fix an index $j \in \{1, \dots, K\}$ and set $r := r_j$.*

*Consider two data windows $\mathbf{X} = \{\mathbf{x}(t)\}_{t=0}^N$ and $\overline{\mathbf{X}} = \{\overline{\mathbf{x}}(t)\}_{t=0}^N$ extracted from the same process equation 1. Suppose $\mathbf{X}$ passes CALIPER's monotone locality test with a positive margin across $\Theta$ and meets the ESS gate at the tightest locality $\theta_{\max}$ (equivalently, at $r_{\min}$), while $\overline{\mathbf{X}}$ fails the same monotone test (also with a positive margin) and meets the ESS gate at $\theta_{\max}$. Then, for any $\delta \in (0, 1)$, there exist constants $\underline{\alpha}, \overline{\alpha} \in (0, 1)$ and $\Delta > 0$ (depending only on $f$, $c$, the noise envelope $\sigma^2$, the grid $\Theta$ (equivalently $\{r_k\}$), the test margins, and the ESS threshold) such that, with probability at least $1 - \delta$,*

$$\inf_{\mathbf{x} \in B} \alpha(\mathbf{x}, r; c) \; \geq \; \underline{\alpha}, \qquad \sup_{\overline{\mathbf{x}} \in B} \alpha(\overline{\mathbf{x}}, r; c) \; \leq \; \overline{\alpha}, \qquad \underline{\alpha} - \overline{\alpha} \; \geq \; \Delta. \tag{3}$$

*In particular, $\mathbf{X}$ exhibits state dependence on $B$ at scale $r$ in the sense of equation 2, whereas $\overline{\mathbf{X}}$ is uniformly less state dependent by a nontrivial margin.*

**Proof sketch of Proposition 1.** Let $\Theta = \{\theta_k\}_{k=1}^K$ be CALIPER's locality grid (equivalently, the induced effective-radius grid $\{r_k\}$), and write $\hat{E}_k(\mathbf{S})$ (empirical) and $E_k(\mathbf{S})$ (population) for the localized one-step error at locality $\theta_k$ (equivalently, at radius $r_k$) on a window $\mathbf{S}$. Because the ESS

gate holds at the tightest locality $\theta_{\max}$ for both $\mathbf{X}$ and $\overline{\mathbf{X}}$, sub-Gaussian concentration under $\beta$-mixing gives the uniform deviation

$$\sup_k |\hat{E}_k(\mathbf{S}) - E_k(\mathbf{S})| \leq \varepsilon_W(\mathbf{S}), \qquad \varepsilon_W(\mathbf{S}) = O\Big(\sqrt{\frac{\log(K/\delta)}{\mathrm{ESS}(\theta_{\max}, \mathbf{S})}}\Big).$$

On $\mathbf{X}$, the monotone test passes with a positive margin: there exists $\tau_{\mathbf{X}} > 0$ such that $\hat{E}_{k+1}(\mathbf{X}) \leq \hat{E}_k(\mathbf{X}) - \tau_{\mathbf{X}}$ for all $k$, hence $E_{k+1}(\mathbf{X}) \leq E_k(\mathbf{X}) - (\tau_{\mathbf{X}} - 2\varepsilon_W(\mathbf{X}))$. Thus shrinking the radius strictly decreases the population localized error. If many pairs with $\|\mathbf{x}' - \mathbf{x}\| \leq r$ violated $\|\mathbf{x}'_+ - \mathbf{x}_+\| \leq cr$, such a decrease could not persist (the deterministic part is absorbed by $c \geq L$), which forces $\inf_{\mathbf{x} \in B} \alpha(\mathbf{x}, r; c) \geq \underline{\alpha}$. Conversely, on $\overline{\mathbf{X}}$ the test fails with a positive margin: there exist $k^\star$ and $\tau_{\overline{\mathbf{X}}} > 0$ such that $\hat{E}_{k^\star+1}(\overline{\mathbf{X}}) \geq \hat{E}_{k^\star}(\overline{\mathbf{X}}) + \tau_{\overline{\mathbf{X}}}$, hence $E_{k^\star+1}(\overline{\mathbf{X}}) \geq E_{k^\star}(\overline{\mathbf{X}}) + (\tau_{\overline{\mathbf{X}}} - 2\varepsilon_W(\overline{\mathbf{X}}))$, which is incompatible with most neighbor pairs remaining $cr$-close after one step; therefore $\sup_{\overline{\mathbf{x}} \in B} \alpha(\overline{\mathbf{x}}, r; c) \leq \overline{\alpha} < \underline{\alpha}$. Taking $\Delta := \underline{\alpha} - \overline{\alpha} > 0$ yields equation 3. $\qquad \square$

**Remark (State dependence and learnability of retraining).** The role of state dependence in retraining can be understood through data-dependent generalization bounds. In particular, results such as (Wei and Ma, 2019) bound the test–train gap for MLP predictors by a term that depends on empirical quantities measured on the training window (e.g., hidden-layer norms and interlayer Jacobian norms). Abstracting these into a single nonnegative complexity term $\mathcal{C}(\mathbf{S})$, one can write (up to logarithmic factors)

$$E_{\text{te}}(h_\psi) - E_{\text{tr}}(h_\psi) \lesssim \mathcal{C}(\mathbf{S})\, n^{-1/2}. \tag{4}$$

Heuristically, when a window is more state dependent on $(B, r)$, radius-$r$ neighbors tend to remain neighbors (after one step) more frequently, so accurate one-step fitting on $B$ can be achieved with less local variation. This tends to reduce empirical Jacobians/norms and hence the data-dependent term $\mathcal{C}(\mathbf{S})$, making the bound equation 4 tighter. Consequently, CALIPER-triggered windows (which indicate stronger state dependence via Proposition 1) are expected to be more favorable for stable retraining on $B$.

**Discussion of assumptions and scope.** The dynamical-systems setting in (1) and the local Lipschitz and $\beta$-mixing conditions are used only for our analysis of CALIPER's trigger, not as requirements of the algorithm itself. In practice, the observed state $\mathbf{x}(t)$ may include a short history of the stream (e.g., via delay embedding), so a first-order Markov model can hold for this augmented state even when the original process depends on $(\mathbf{x}(t - k), \ldots, \mathbf{x}(t))$. Our distance-based neighborhoods operate on normalized features and are guarded by the $\mathrm{ESS}(\theta_{\max}) \geq C(d + 1)$ gate, so CALIPER naturally asks for larger windows in higher dimensions; when the intrinsic dimension is very large, combining CALIPER with standard dimensionality reduction is sensible. Overall, we treat these as regularity assumptions for the theory, while the algorithm itself applies more broadly, as illustrated by our chaotic and noisy benchmarks.

## 3    EXPERIMENTAL RESULTS

We evaluate CALIPER through experiments designed to answer three questions: (Q1) **Effectiveness**—how accurately does CALIPER estimate the data required to retrain a model after a detected drift? (Q2) **Scalability**—what is the computational overhead of CALIPER as data increase? (Q3) **Adaptation**—under sudden drift, does retraining with CALIPER outperform incremental updates?

**Datasets.** We use four datasets from different domains: (a) **MoCap**, sequences from the CMU Motion Capture Database[1]; (b) **TEP**, the Tennessee Eastman Process—a benchmark discrete-time simulation of a chemical plant (Downs and Vogel, 1993); (c) **Automobile**, five synchronized vehicle sensors (accelerometer, speed, $G_x$, $G_y$, $G_z$) across multiple driving courses; and (d) **Dysts**, time series from the DYSTS library (Gilpin, 2021) covering chaotic systems with known dynamical properties. Experimental settings are detailed in Appendix F.

**Experimental Setup.** The experimental framework employed multiple algorithms and drift detectors. The base learners included kernel ridge regression (KRR), MLP, and Transformer. We also used ADWIN (Bifet and Gavaldà, 2007) and KSWIN (Raab et al., 2020) for drift detection. Performance

---

[1]http://mocap.cs.cmu.edu/

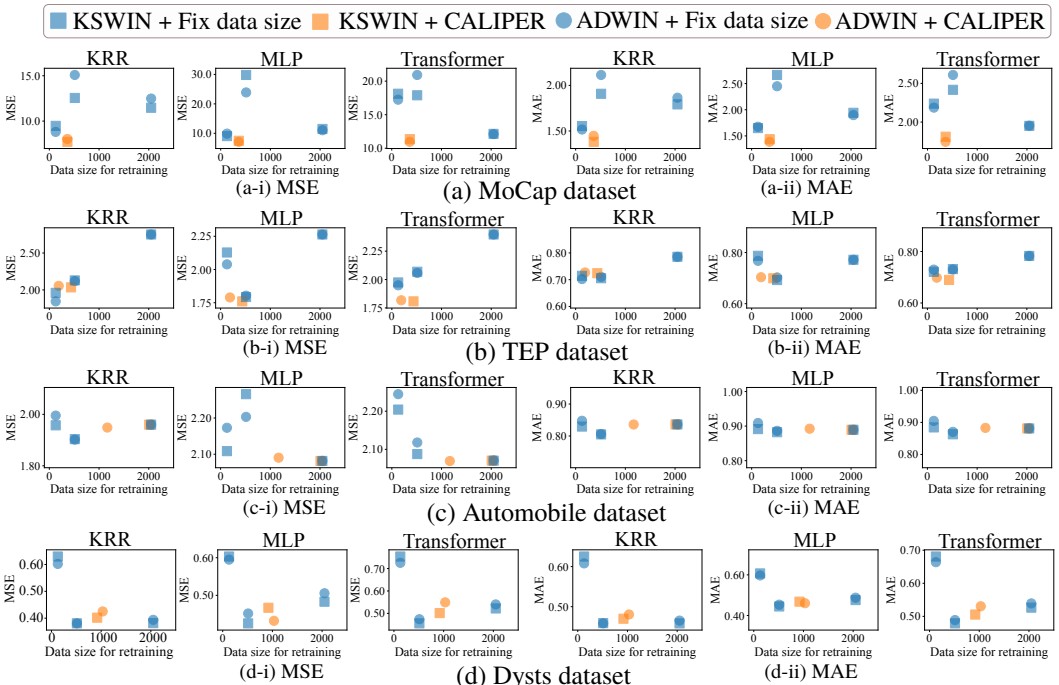

Figure 3: **Performance of CALIPER on four datasets (MoCap, TEP, Automobile, Dysts) and three model families (KRR, MLP, Transformer). We compare fixed data sizes (128/512/2048; blue) with CALIPER (orange, "CALIPER"). Each panel reports MSE (left, "-i") and MAE (right, "-ii") as a function of the retraining data size after each drift detected by ADWIN (circles) or KSWIN (squares). CALIPER matches or exceeds the best fixed data size without per-dataset tuning; notably, the data size selected by CALIPER typically aligns with the dataset-specific optimal fixed data size. See Tables 3 and 2 for full results in Appendix G.**

metrics included prediction mean squared error (MSE) and mean absolute error (MAE). Each dataset was subjected to multiple independent runs with random seeds, and all algorithms within each run shared the same random seed initialization. Our experimental settings are detailed in Appendix F.

## (Q1) EFFECTIVENESS

Fig. 3 compares fixed data sizes (128/512/2048; blue) with CALIPER ("CALIPER", orange) across four datasets (MoCap, TEP, Automobile, Dysts), three model families (KRR, MLP, Transformer), two drift detectors (ADWIN/KSWIN; circles/squares), and two metrics (MSE "-i", MAE "-ii"). Each panel plots error as a function of the retraining data size used after each detected drift; for CALIPER, the x-value is the average data size consumed per retraining. Across datasets, detectors, and architectures, CALIPER sits near the best fixed point, matching or exceeding the prediction accuracy of the strongest fixed data size without per dataset tuning. On MoCap, it consistently attains the panel-wise minimum; on TEP, Automobile, and Dysts, it remains competitive with the best fixed choice. Overall, selecting the data size materially impacts accuracy, and CALIPER provides near-optimal choices across conditions. Even when a fixed data size numerically wins, that merely shows ex post that it happened to be optimal—not that we can know the stream-time choice to be optimal at the time. Moreover, the fixed size with the best prediction accuracy varies widely. The fixed size with the best prediction accuracy varies widely. For example, the MLP model achieves its highest accuracy on TEP with a data size of 512, whereas the same setting yields the lowest performance on MoCap. These results underscore the brittleness of a priori data size selection. In contrast, CALIPER selects data sizes data-dependently near the optimum and typically achieves best- or second-best accuracy, thereby preserving the method's practical value. For a tree-based learner, CALIPER exhibits the same qualitative behavior, with estimated window sizes remaining close to empirically optimal choices. These tree-based results, together with full numerical results averaged over horizons 1, 15, and 30 (Tables 3 and 2) and a hyperparameter-sensitivity study of CALIPER's

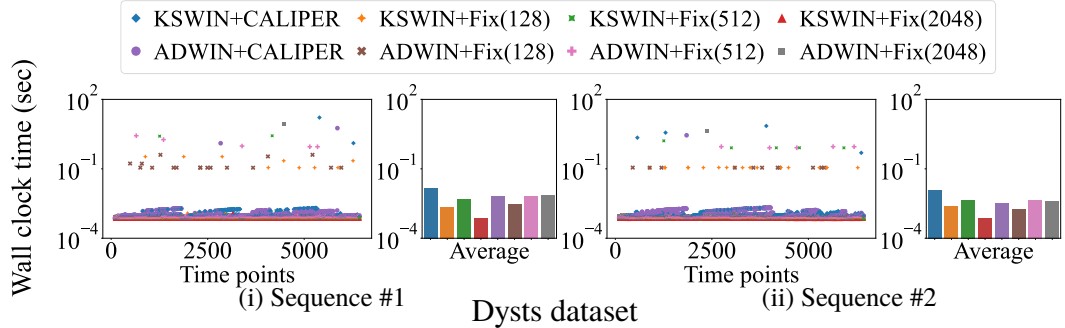

**Figure 4: Per time step and average wall clock time on Dysts sequences #1 and #2 for AD-WIN/KSWIN with CALIPER and fixed-size buffers (128/512/2048); curves are flat with low means, and occasional spikes reflect retraining rather than CALIPER.**

Table 1: Adaptation after drift. CALIPER-triggered retraining vs. Incremental updating. MSE/MAE averaged over (1,15,30) with the past sequence length is 30; detectors ADWIN/KSWIN; models KRR/MLP/Transformer. Lower is better; best bold, second underlined. Tables 3 and 2 in the Appendix for the full results.

| Model | Detector | MoCap | | TEP | | Automobile | | Dysts | |
|-------|----------|-------|-------|-------|-------|------------|-------|-------|-------|
| | | MSE | MAE | MSE | MAE | MSE | MAE | MSE | MAE |
| MLP | ADWIN | **7.106** | **1.387** | 1.790 | 0.704 | 2.090 | 0.892 | **0.432** | **0.460** |
| | KSWIN | 7.449 | 1.436 | **1.760** | **0.699** | **2.081** | **0.889** | 0.467 | 0.468 |
| | (Incremental) | 412.6 | 8.462 | 3.699 | 0.879 | 2.406 | 0.894 | 71.75 | 1.524 |
| Transformer | ADWIN | **10.96** | **1.744** | 1.818 | 0.698 | 1.948 | 0.883 | 0.549 | 0.530 |
| | KSWIN | 11.35 | 1.810 | **1.808** | **0.690** | 2.070 | 0.881 | **0.501** | **0.505** |
| | (Incremental) | 22.08 | 2.654 | 2.376 | 0.767 | **1.866** | **0.773** | 0.582 | 0.585 |

locality grid $\theta$ and ESS threshold $C$ for the MLP base learner (showing that selected window sizes and post-drift errors are stable over a wide range of settings), are all reported in Appendix G.

### (Q2) SCALABILITY

We measure wall clock time on the Dysts dataset using two sequences (#1, #2) and eight detector–strategy combinations (ADWIN/KSWIN paired with either CALIPER or fixed data size of 128/512/2048), with an MLP as the base learner. Fig. 4 reports per time step wall clock time (log-seconds) along with the corresponding averages. Across all combinations, the wall clock time curves are essentially flat and the mean cost per step is small; occasional spikes align with model retraining and are not caused by CALIPER. Overall, this indicates that CALIPER adds negligible overhead relative to the base learner, the detector, and the fixed-length baseline, consistent with our use of lightweight local regression. Additional results appear in Appendix G.

### (Q3) ADAPTATION

Table 1 compares CALIPER with an incremental-update baseline (online SGD; the same architecture, no explicit retraining). Results are averaged over horizons $(1, 15, 30)$ with a past sequence length of 30. Overall, CALIPER matches or surpasses incremental updates across datasets and models. For **MLP**, CALIPER yields large gains on MoCap and Dysts (e.g., MoCap: MSE 7.106 vs. 412.6; Dysts: 0.432 vs. 71.75), indicating that purely local incremental steps can be unstable under drift. For **Transformer**, CALIPER is competitive on all datasets and superior on TEP and Dysts (e.g., Dysts: MSE 0.501 with KSWIN vs. 0.582 incremental); on Automobile, incremental performs slightly better (MSE 1.866 vs. 1.948–2.070), but the gap is modest. Taken together, these results suggest that (a) selecting an appropriate retraining data size at each drift materially improves accuracy and stability, and (b) pure incremental updates are often insufficient in sudden drift.

## 4 RELATED WORK

### 4.1 CONCEPT DRIFT

In dynamically changing and non-stationary environments, the data distribution can change over time, yielding the phenomenon of concept drift. Concept drift is a phenomenon in which the statistical properties of a region of interest change over time in an arbitrary manner (Lu et al., 2014). It was first proposed by (Schlimmer and Granger, 1986), who pointed out that noise data can become non-noise information at different times. Such changes could be caused by changes in hidden variables (Liu et al., 2017) that cannot be measured directly. Strategies for updating existing training models in response to the drift caused by such system changes can be categorized into two main groups: retraining and model adjustment. Each of these aims to address different types of drift. Here, we primarily focus on the retraining analysis aspect and summarize the model adjustment based approaches in Appendix B. Window-based detectors (e.g., ADWIN, KSWIN) are standard in streaming settings, where they compare recent and historical windows to flag drift (Bifet and Gavaldà, 2007; Raab et al., 2020); see the introduction for a brief survey. Crucially, these detectors respond to drift but not how much post-drift data are required for reliable retraining—a gap our work addresses with a post-drift data sufficiency criterion.

### 4.2 ANALYSIS OF NONLINEAR DYNAMICS

Inspired by Wold's theorem (Wold, 1938), time series data $X_t$ can be formally decomposed as $X_t = \sum_{j=0}^{\infty} b_j \epsilon_{t-j} + \eta_t$. Here, $\eta_t$ represents the deterministic component, and $\epsilon_t$ represents the stochastic component as a stationary process input to the linear filter $b_j$. In many tasks—especially time-series forecasting—the deterministic part is crucial: long-horizon trends capture low-frequency evolution. This deterministic structure is often modeled with flexible function classes, including programmatically discovered structures (Champion et al., 2019; Zheng et al., 2019; Bertsimas and Gurnee, 2023; Fujiwara et al., 2025) and analyzed for description and prediction. A central phenomenon in nonlinear dynamics is state dependence (Sugihara, 1994; Ye et al., 2015), which measures how similarly futures unfold when present states are similar; operationally, a series exhibits state dependence if forecasts from a model trained locally around a point $x$ surpass those from a global model trained on all data. This property underpins short-term prediction for nonlinear series, exemplified by S-Map (Hsieh et al., 2005; Perretti et al., 2013; Deyle et al., 2016; Ushio et al., 2018), which computes distances between a target state and a historical library and performs distance-weighted regression for forecasting. While such analyses rely on mild assumptions, a key advantage is their independence from a posited generative model. Building on this insight, our work repurposes state dependence to assess data sufficiency. Beyond offering a model-agnostic criterion that estimates the minimum data requirement without requiring knowledge of the parametric system, it also provides a practical estimation algorithm that enables deployment in real-world settings.

## 5 CONCLUSION

In this paper, we introduced CALIPER, a detector- and model-agnostic, data-only framework that returns the earliest post-drift window that satisfies a data-side stopping rule, used as a proxy for retraining readiness. Rather than probing or stress-testing a downstream model, CALIPER enforces a simple, verifiable stopping rule on the stream itself: it checks that local neighborhoods are sufficiently populated (via an effective sample size gate) and that one-step predictability improves monotonically as neighborhoods become more local, all computed in a single pass using low-overhead weighted local regressions. Our theory shows that this trigger is not merely heuristic: under a stylized dynamical model, passing it implies stronger state dependence; under standard regularity assumptions, this can be favorable for learnability on a suitable local region—providing a principled basis for post-drift sample sizing and for deciding when enough data has accumulated to retrain safely. Empirically, across four datasets, three learner families, and two drift detectors, CALIPER matches or surpasses the best fixed-size retraining without per-dataset tuning, reduces post-drift error, and consistently outperforms incremental updates at negligible computational and memory cost. In practice, CALIPER cleanly separates the when from the how of adaptation, enabling plug-and-play deployment in streaming systems with heterogeneous models and scarce labels, and making retraining decisions transparent, auditable, and robust to detector choice.

ACKNOWLEDGMENTS

This work was supported by JSPS KAKENHI Grant-in-Aid for Scientific Research Number JP24KJ1618, JST CREST JPMJCR23M3, JST START JPMJST2553, JST CREST JPMJCR20C6, JST K Program JPMJKP25Y6, JST COI-NEXT JPMJPF2009, JST COI-NEXT JPMJPF2115, the Future Social Value Co-Creation Project - Osaka University.

ETHICS STATEMENT

We adhere to the ICLR Code of Ethics. Some experiments use a closed automobile dataset that may contain individual driving records provided under a data-use agreement. No raw personally identifiable information is shared in this paper or the supplementary materials; access is restricted to authorized researchers, and analysis is conducted on de-identified records following the provider's privacy policies. The dataset cannot be redistributed. We disclose no conflicts of interest.

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

TECHNICAL APPENDICES AND SUPPLEMENTARY MATERIAL

## A  ALGORITHM OVERVIEW

In this section, we provide the details of the CALIPER optimization algorithm proposed in section 2. Algorithm 1 provides an overview of CALIPER.

---

**Algorithm 1** CALIPER $(\mathbf{X}_t, E_{t-1})$

---

**Input:** Post-drift window: $\mathbf{X}_t = [\mathbf{x}(t_s), ..., \mathbf{x}(t)]$ and previous prediction error at each $\theta$: $E_{t-1}$
**Output:** Retrain flag and prediction error at each $\theta$: $E_t$
 1:
 2: /* (i) Window Normalization and Split. */
 3: $\mathbf{Z} \leftarrow$ normalized $\mathbf{X}_t$
 4: $\mathbf{X}_h = \mathbf{Z}[1 : n_t - 2]$; $\mathbf{Y}_h = \mathbf{Z}[2 : n_t - 1]$; $\mathbf{x}_q = \mathbf{Z}[n_t - 1]$; $\mathbf{y}_q = \mathbf{Z}[n_t]$;
 5:
 6: /* (ii) ESS Check. */
 7: **for all** $\mathbf{x}_i \in \mathbf{X}_h$ **do**
 8:     $r_i^{raw} = \|\mathbf{x}_i - \mathbf{x}_q\|$
 9: **end for**
10: $\mathbf{r} = \mathbf{r}^{\mathbf{raw}}/\text{mean}(\mathbf{r}^{\mathbf{raw}})$
11: **for all** $\mathbf{x}_i \in \mathbf{X}_h$ **do**
12:     $w_i = \exp(-\theta_{max} * r_i)$
13: **end for**
14: **if** $(\sum w_i)^2/\sum w_i^2 < C * (d + 1)$ **then**
15:     **return false**, $E_{t-1}$
16: **end if**
17:
18: /* (iii) Weighted Local Regression */
19: **for all** $\theta \in [0, .., \theta_{max}]$ **do**
20:     $\mathbf{w} = \exp(-\theta * \mathbf{r})$
21:     Learn weighted local regression WLR$_\theta$ using $\{\mathbf{X}_h, \mathbf{Y}_h\}$ and $\mathbf{w}$.
22:     $e_{(t,\theta)} = \|\mathbf{y}_q - \text{WLR}_\theta(\mathbf{x}_q)\|$
23:     $E_{(t,\theta)} = E_{(t-1,\theta)} + e_{(t,\theta)}$
24: **end for**
25:
26: /* (iv) Test & Trigger. */
27: $\mathbf{E}_t = \{E_{(t,\theta)}\}_{\theta=0}^{\theta^{max}}$
28: **if** $\mathbf{E}_t$ is monotonically non-increasing for $\theta$ **then**
29:     **return true**, $E_t$
30: **else**
31:     **return false**, $E_t$
32: **end if**

---

## B  ADDITIONAL RELATED WORKS

Instead of retraining the entire model, there are ways to develop models that adaptively learn from changing data. This approach is more efficient than retraining if the drift occurs only in localized regions. This approach can be easily implemented by stochastic gradient descent, and various methods have been proposed in recent years. In addition to standard fine-tuning techniques, recent studies (Pham et al., 2023; Zhang et al., 2023; Zhao and Shen, 2025) have proposed more sophisticated model adaptation techniques. These focus on ways to effectively adapt to recent data by using predictive feedback (e.g., errors and gradients) on new training samples. Among them, FSNet (Pham et al., 2023) monitors the gradients in previous fine-tunings and translates them into parameter adjustments to adapt the new predictive model to the current training sample, and dynamically adjusts the ensemble weights and the model's parameter weights according to the prediction error. OneNet (Zhang et al., 2023) is an online ensemble network that generates ensemble weights to combine prediction models and dynamically adjusts ensemble weights and model parameter weights according to prediction error. PROCEED (Zhao and Shen, 2025) is a framework that estimates the drift between recent training samples and the current test sample. It then proactively adjusts the model parameters before receiving

ground-truth future values. This approach effectively bridges the temporal gap caused by forecast horizon delays.

## C    Limitation and Future work

Our theoretical analysis adopts a stylized but standard one-step state-space model,

$$s(t+1) = f(s(t)) + \xi_t,$$

with locally Lipschitz $f$ and sub-Gaussian noise, to formalize state dependence and to provide an interpretation relating CALIPER's trigger (a monotone locality curve plus an ESS condition) to learnability on a suitable local region. These assumptions should be interpreted as regularity conditions for the analysis, not as hard requirements for applying the algorithm. In practice, CALIPER operates on arbitrary feature vectors, and the state can include a short history of the stream (e.g., via delay embedding) or a representation learned by an upstream encoder. Nevertheless, our current guarantees do not explicitly cover fully non-Markovian dynamics or settings with strong latent or exogenous drivers beyond what is captured by the chosen representation. Extending the formal results to explicit latent-variable and history-dependent models is an important direction for future work.

A second limitation concerns high-dimensional geometry. CALIPER relies on distance-based local neighborhoods to form localized fits, and naive nearest-neighbor weighting can be unreliable in very high dimensions: neighborhoods become sparse, effective sample sizes can collapse, and the localized regression can degrade unless the post-drift window is sufficiently large. CALIPER mitigates this via (i) feature/distance normalization within the current post-drift window and (ii) the ESS gate as a conservative safeguard (checked at the tightest locality), which naturally forces larger windows as dimensionality increases. However, this does not eliminate the fundamental dependence on having a meaningful metric in the working representation space. For extremely high-dimensional raw inputs (e.g., image streams), compact representations or adaptive metrics are likely necessary; characterizing when such representations yield reliable locality and ESS behavior remains future work.

Finally, CALIPER should be interpreted as detecting a regime in which a broad class of reasonably expressive learners can retrain stably, rather than identifying a single universal optimum window. Establishing a tighter theoretical link between the trigger and model-specific convergence is an interesting direction for future work.

## D    The Use of Large Language Models (LLMs)

We use LLMs to aid or polish writing. Specifically, we used LLMs for assistance when writing papers, for example to check spelling and to make grammar suggestions.

## E    Proof of Proposition 1

*Proof of Proposition 1.* We work under equation 1 and equation 2 with a fixed compact $B \subset \mathbb{R}^d$, scale $r > 0$, and $c \geq L$. Let $\Theta = \{\theta_k\}_{k=1}^K$ be CALIPER's locality grid, ordered so that $0 < \theta_1 < \cdots < \theta_K = \theta_{\max}$. Fix $\tau \in (0,1)$ and define the induced effective-radius grid by

$$r_k := r^{\mathrm{eff}}(\theta_k; \tau) = \frac{\log(1/\tau)}{\theta_k}, \qquad k = 1, \ldots, K,$$

so that increasing $\theta_k$ corresponds to tighter locality (smaller $r_k$). We write $\hat{E}_k(\mathbf{S})$ and $E_k(\mathbf{S})$ for the empirical and population localized one-step errors at locality $\theta_k$ (equivalently, at radius $r_k$) on window $\mathbf{S}$.

Step 1 (Uniform concentration). By sub-Gaussian coordinates, $\beta$-mixing with summable coefficients, and the ESS gate at the tightest locality $\theta_{\max}$, there exists, for any $\delta \in (0,1)$ and with probability at least $1 - \delta$, a uniform deviation bound

$$\sup_{1 \leq k \leq K} |\hat{E}_k(\mathbf{S}) - E_k(\mathbf{S})| \leq \varepsilon_W(\mathbf{S}), \qquad \varepsilon_W(\mathbf{S}) = O\left(\sqrt{\frac{\log(K/\delta)}{\mathrm{ESS}(\theta_{\max}, \mathbf{S})}}\right), \quad \mathbf{S} \in \{\mathbf{X}, \overline{\mathbf{X}}\}.$$

Step 2 (Triggered window $\mathbf{X}$: lower bound on $\alpha$). On $\mathbf{X}$, the empirical monotone test passes across the grid with a positive margin; hence there exists $\tau_{\mathbf{X}} > 0$ such that

$$\hat{E}_{k+1}(\mathbf{X}) \ \leq \ \hat{E}_k(\mathbf{X}) - \tau_{\mathbf{X}} \qquad (k = 1, \ldots, K-1).$$

By Step 1,

$$E_{k+1}(\mathbf{X}) \ \leq \ E_k(\mathbf{X}) - (\tau_{\mathbf{X}} - 2\varepsilon_W(\mathbf{X})) \qquad (k = 1, \ldots, K-1),$$

with probability at least $1 - \delta$, and the gate at $\theta_{\max}$ ensures $\tau_{\mathbf{X}} - 2\varepsilon_W(\mathbf{X}) > 0$. If $\inf_{\mathbf{x} \in B} \alpha(\mathbf{x}, r; c)$ were too small, then a nonnegligible fraction of radius-$r$ neighbor pairs would satisfy $\|\overline{\mathbf{x}}_+ - \mathbf{x}_+\| > cr$, which—after absorbing the deterministic Lipschitz part through $c \geq L$—would prevent $E_{k+1}(\mathbf{X}) < E_k(\mathbf{X})$ from holding uniformly as the radius shrinks, a contradiction. Therefore there exists $\underline{\alpha} \in (0,1)$ such that

$$\inf_{\mathbf{x} \in B} \alpha(\mathbf{x}, r; c) \ \geq \ \underline{\alpha} \quad \text{on } \mathbf{X}$$

with probability at least $1 - \delta$.

Step 3 (Non-triggered window $\overline{\mathbf{X}}$: upper bound on $\alpha$ and gap). On $\overline{\mathbf{X}}$, the empirical monotone test fails with a positive margin, so there exist $k^\star$ and $\tau_{\overline{\mathbf{X}}} > 0$ such that

$$\hat{E}_{k^\star+1}(\overline{\mathbf{X}}) \ \geq \ \hat{E}_{k^\star}(\overline{\mathbf{X}}) + \tau_{\overline{\mathbf{X}}}.$$

By Step 1,

$$E_{k^\star+1}(\overline{\mathbf{X}}) \ \geq \ E_{k^\star}(\overline{\mathbf{X}}) + (\tau_{\overline{\mathbf{X}}} - 2\varepsilon_W(\overline{\mathbf{X}})),$$

with probability at least $1 - \delta$ (and we ensure $\tau_{\overline{\mathbf{X}}} - 2\varepsilon_W(\overline{\mathbf{X}}) > 0$ via the gate at $\theta_{\max}$). If $\sup_{\overline{\mathbf{x}} \in B} \alpha(\overline{\mathbf{x}}, r; c)$ were close to one, shrinking the radius would not increase the population localized error, contradicting the display. Hence there exists $\overline{\alpha} \in (0,1)$ such that

$$\sup_{\overline{\mathbf{x}} \in B} \alpha(\overline{\mathbf{x}}, r; c) \ \leq \ \overline{\alpha} \quad \text{on } \overline{\mathbf{X}}$$

with probability at least $1 - \delta$. Setting $\Delta := \underline{\alpha} - \overline{\alpha} > 0$ yields equation 3 and completes the proof. $\square$

## F  EXPERIMENTAL SETTINGS

**Code.** Our datasets and source code are available at: https://github.com/renfujiwara/CALIPER

**Hyperparameters of CALIPER.** In CALIPER, $\theta$ is selected from $[0, 0.1, 1.0, 2.0, 4.0, 8.0, 16.0]$ and we set $C = 3$. Then, it is determined whether the error non-increases monotonically with respect to $\theta$.

**Computing infrastructure.** The configuration includes 2 * Xeon Gold 6444Y 3.6GHz CPU, 12 * 64GB DDR4 RAM (768GB), and NVIDIA RTX A6000 48GB GPU, which is sufficient for all the baselines.

**Dysts dataset details.** These data are obtained from dysts database (Gilpin, 2021), which provides data, equations, and dynamical properties for chaotic systems exhibiting strange attractors and coming from disparate scientific fields. In our experiments, we consider 12 hyperchaotic systems representing ordinary differential equations (ODEs) with polynomial nonlinearities. We used 10 systems for training and tested them with the remaining systems. Each system is included in the test data at least twice. In other words, we conducted our experiments with 12 different synthetic data sets (#1-#12). We use 1500 lengths of data for the warm-up phase and 6000 lengths of data that we generated for the online learning period. The time step between each sample is 0.05, and the initial conditions are randomly generated according to (Gilpin, 2021).

**Implementation Details.** In the MoCap, Automobile, and Dysts datasets, we split the data into warm-up and online-training phases with a 20:80 ratio. We used the TEP dataset, where the normal operating interval was employed as the warm-up phase, and the abnormal interval was treated as the online-training phase. Following (Zhou et al., 2021), we minimize the mean squared error (MSE) using the AdamW optimizer (Loshchilov and Hutter, 2019); the learning rates for both the MLP and Transformer were selected from $\{10^{-3}, 5 \times 10^{-3}, 10^{-4}, 5 \times 10^{-4}, 10^{-5}\}$ based on

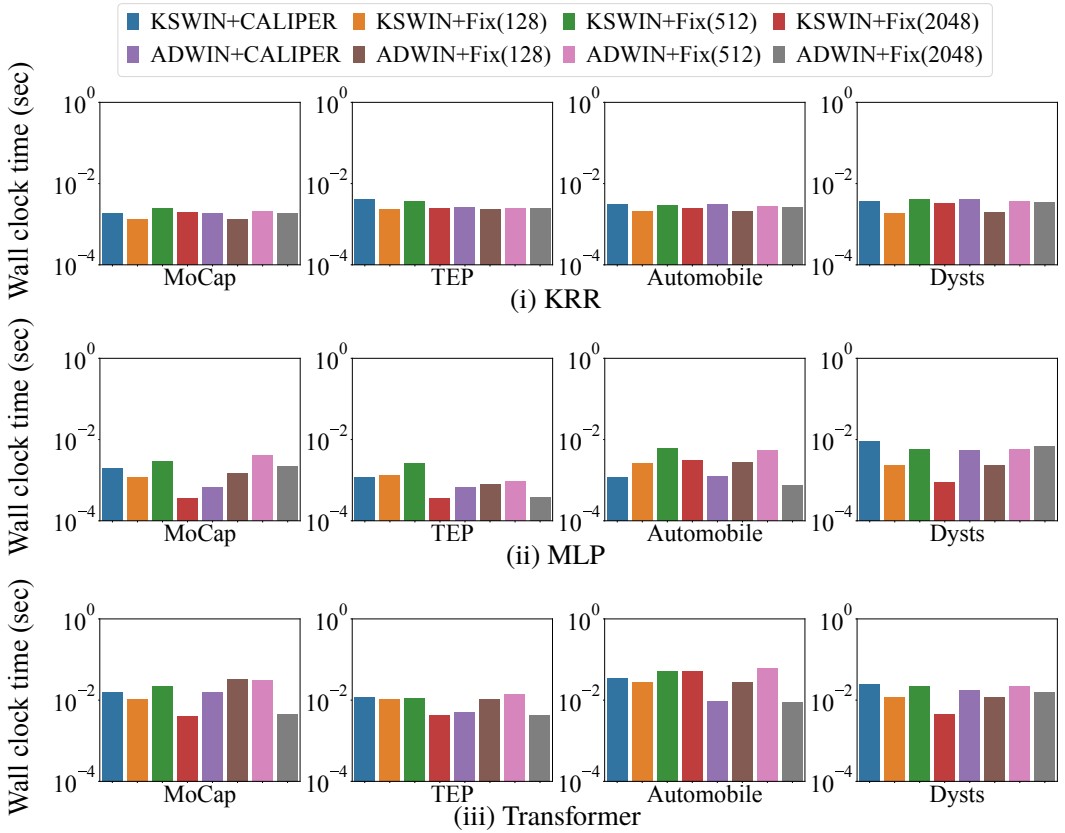

**Figure 5: The average per time step wall-clock time (bar chart) for MoCap, TEP, Automobile, and Dysts under ADWIN/KSWIN with CALIPER vs. fixed data size (128/512/2048) using KRR/MLP/Transformer; CALIPER matches fixed data size baselines.**

validation performance within the warm-up phase. The MLP uses a hidden dimension of 32, and the Transformer hyperparameters and other training settings follow the benchmark implementation.[2] For KRR, we implemented scikit-learn's `KernelRidge` (Pedregosa et al., 2011) with an RBF kernel and search over $\gamma \in \{\text{None}, 10^{-3}, 10^{-2}, 10^{-1}, 1.0\}$ and $\alpha \in \{10^{-2}, 10^{-1}, 1.0, 10.0\}$, selecting the combination based on the validation performance within the warm-up phase. For drift detection, we used ADWIN with a confidence value of 0.002 and KSWIN with a significance level of 0.05.

# G    ADDITIONAL EXPERIMENTAL RESULTS

## G.1    ADDITIONAL ANALYSES

**Hyperparameter sensitivity of CALIPER.** Figure 6 shows the sensitivity of CALIPER to its two main hyperparameters: the locality parameter $\theta_{\max}$ and the ESS multiplier $C$. For each dataset and each drift detector (ADWIN, KSWIN), we vary $C \in \{2, 3, 4\}$ and $\theta_{\max} \in \{12, 14, 16, 18, 20\}$ while keeping all other settings fixed, and plot the resulting test MAE. The y-axis of Fig. 6 uses the same absolute MAE scale as our main results. On three of the four datasets, the curves are almost flat for both hyperparameters, indicating that CALIPER is highly robust in these regimes. On the more challenging MoCap dataset, changing $C$ still has a limited effect, whereas large values of $\theta_{\max}$ (around 18–20) lead to performance degradation. This behaviour agrees with the interpretation of $\theta_{\max}$ as a locality scale: if the window becomes too wide, the WLR probe no longer reflects local state dependence and the retraining trigger deteriorates. Around the default configuration ($\theta_{\max} = 16$, $C = 3$) used in all other experiments, the MAE remains close to the optimum on all datasets.

---

[2]`https://github.com/thuml/Time-Series-Library`

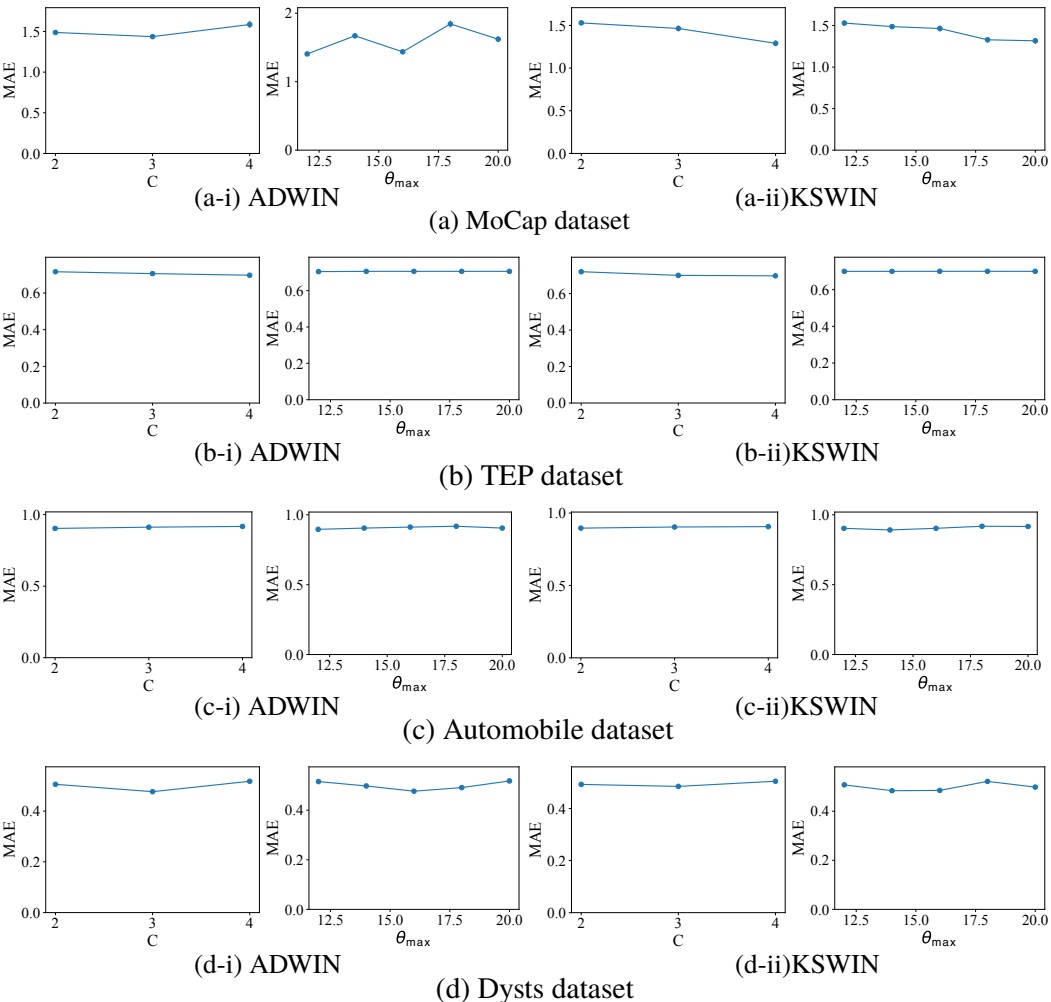

**Figure 6: Hyperparameter sensitivity of CALIPER. Test MAE as a function of the ESS multiplier** $C \in \{2, 3, 4\}$ **(left panels) and the locality parameter** $\theta_{\max} \in \{12, 14, 16, 18, 20\}$ **(right panels) on the four datasets and for both ADWIN and KSWIN. The y-axis is on the same absolute MAE scale as in the main results. The curves are nearly flat on three of the datasets, and only large values of** $\theta_{\max}$ **on the MoCap dataset cause a noticeable degradation, consistent with the role of** $\theta_{\max}$ **as a locality parameter.**

**Full Experimental Results for Prediction.** Tables 2 and 3 show full numerical results (averaged over horizons (1,15,30)). As discussed in Section 3, our method achieves performance that is comparable to or even substantially better than fixed-length retraining strategies, despite not having access to the final predictive accuracy in advance. A noteworthy observation is that the optimal fixed data size window varies across datasets. For instance, smaller data sizes are more advantageous for the MoCap and TEP datasets, whereas larger data sizes tend to perform better for the others. This highlights the difficulty of relying on a fixed data size retraining strategy and underscores the effectiveness of our approach.

**Tree-based learner (ExtraTrees).** We also include an additional experiment using an Extremely Randomized Trees (ExtraTrees) regressor as a tree-based base learner. We keep the same detectors (ADWIN and KSWIN), datasets, and CALIPER hyperparameters as in the main experiments. For each dataset and detector, we sweep several fixed post-drift window sizes and compare them to CALIPER. Across all settings, CALIPER's estimated window sizes remain close to the empirically optimal fixed choice, and its post-drift errors match or improve upon the best fixed window. These results confirm that CALIPER behaves similarly for tree-based models as for KRR, MLP, and

Transformers. Representative curves and full numerical values (averaged over horizons 1, 15, and 30) are reported in Tables 2 and 3.

**Additional Scalability Results.** Figure 5 shows dataset-level average wall clock time per time step (bar chart) across four datasets (MoCap, TEP, Automobile, Dysts), two detectors (ADWIN/KSWIN), and two strategies (CALIPER vs. fixed data size of 128/512/2048) under three base learners (KRR, MLP, Transformer). Across all settings, CALIPER is on par with the fixed data size baselines, indicating negligible additional overhead; remaining differences are largely explained by the base learner and detector rather than the strategy.

Table 2: Full results with ADWIN. For each dataset and model family, we report MSE/MAE at prediction horizons (1, 15, 30) and their average. We compare CALIPER (ours) with fixed retraining windows of 128, 512, and 2048 steps ("Fix (·)") and with an online Incremental baseline (no explicit window retraining). The past sequence length is 30. Datasets span four heterogeneous domains (TEP, MoCap, Automobile, Dysts). Lower is better; best per column in bold (second best underlined, if applicable).

| Dataset | Model | $l_s$ | CALIPER MSE | CALIPER MAE | Fix (128) MSE | Fix (128) MAE | Fix (512) MSE | Fix (512) MAE | Fix (2048) MSE | Fix (2048) MAE | Incremental MSE | Incremental MAE |
|---|---|---|---|---|---|---|---|---|---|---|---|---|
| MoCap | KRR | 1 | **6.952** | **1.146** | 7.607 | 1.339 | 12.369 | 1.719 | 8.074 | 1.228 | 16.003 | 2.201 |
| | | 15 | **7.691** | **1.480** | 8.934 | 1.561 | 15.165 | 2.230 | 10.010 | 1.802 | 19.552 | 2.538 |
| | | 30 | **9.431** | 1.716 | 9.879 | **1.650** | 17.725 | 2.395 | 19.373 | 2.565 | 18.946 | 2.524 |
| | | Avg | **8.025** | **1.447** | 8.807 | 1.516 | 15.086 | 2.114 | 12.486 | 1.865 | 18.167 | 2.421 |
| | ExtraTrees | 1 | **7.822** | **1.455** | 15.295 | 2.025 | 11.100 | 1.804 | 9.541 | 1.626 | 35.622 | 3.337 |
| | | 15 | **9.846** | **1.701** | 17.791 | 2.198 | 13.351 | 2.030 | 10.599 | 1.806 | 44.145 | 3.777 |
| | | 30 | **11.127** | **1.825** | 15.504 | 2.082 | 13.479 | 2.028 | 11.610 | 1.910 | 33.087 | 3.298 |
| | | Avg | **9.598** | **1.660** | 16.196 | 2.102 | 12.643 | 1.954 | 10.584 | 1.781 | 37.618 | 3.471 |
| | MLP | 1 | 3.435 | 0.966 | 6.169 | 1.361 | 5.548 | 1.409 | **2.199** | **0.919** | 357.793 | 7.636 |
| | | 15 | **7.534** | **1.469** | 9.392 | 1.705 | 24.581 | 2.570 | 9.661 | 1.928 | 578.383 | 9.466 |
| | | 30 | **10.349** | **1.725** | 14.122 | 1.959 | 41.531 | 3.368 | 21.282 | 2.845 | 301.643 | 8.285 |
| | | Avg | **7.106** | **1.387** | 9.894 | 1.675 | 23.887 | 2.449 | 11.047 | 1.897 | 412.606 | 8.462 |
| | Transformer | 1 | **9.458** | **1.571** | 14.566 | 2.013 | 17.245 | 2.324 | 10.666 | 1.779 | 19.661 | 2.508 |
| | | 15 | **10.134** | **1.681** | 17.627 | 2.185 | 20.548 | 2.581 | 12.353 | 1.988 | 22.874 | 2.708 |
| | | 30 | **13.273** | **1.981** | 19.514 | 2.361 | 24.971 | 2.922 | 13.308 | 2.093 | 23.700 | 2.748 |
| | | Avg | **10.955** | **1.744** | 17.236 | 2.186 | 20.921 | 2.609 | 12.109 | 1.953 | 22.078 | 2.654 |
| TEP | KRR | 1 | 1.943 | 0.690 | 1.696 | **0.669** | 1.952 | 0.669 | 2.638 | 0.749 | **1.671** | 0.691 |
| | | 15 | 2.055 | 0.731 | 1.861 | **0.707** | 2.134 | 0.716 | 2.752 | 0.789 | **1.794** | 0.708 |
| | | 30 | 2.166 | 0.760 | 1.977 | 0.732 | 2.275 | 0.746 | 2.863 | 0.819 | **1.874** | **0.718** |
| | | Avg | 2.055 | 0.727 | 1.845 | **0.702** | 2.120 | 0.710 | 2.751 | 0.786 | **1.780** | 0.706 |
| | ExtraTrees | 1 | 1.674 | 0.637 | 1.383 | 0.638 | 1.820 | 0.667 | 2.340 | 0.734 | **1.314** | **0.621** |
| | | 15 | 1.827 | 0.685 | 1.682 | 0.697 | 2.174 | 0.731 | 2.749 | 0.800 | **1.572** | **0.663** |
| | | 30 | 1.951 | 0.721 | 1.883 | 0.728 | 2.411 | 0.770 | 3.007 | 0.840 | **1.716** | **0.678** |
| | | Avg | 1.817 | 0.681 | 1.649 | 0.688 | 2.135 | 0.722 | 2.699 | 0.792 | **1.534** | **0.654** |
| | MLP | 1 | 1.651 | 0.663 | 1.905 | 0.746 | **1.478** | **0.642** | 1.910 | 0.706 | 3.136 | 0.829 |
| | | 15 | 1.739 | 0.702 | 2.051 | 0.765 | **1.684** | **0.698** | 2.044 | 0.760 | 4.180 | 0.914 |
| | | 30 | **1.979** | **0.746** | 2.161 | 0.792 | 2.247 | 0.769 | 2.837 | 0.849 | 3.781 | 0.894 |
| | | Avg | **1.790** | 0.704 | 2.039 | 0.768 | 1.803 | **0.703** | 2.264 | 0.772 | 3.699 | 0.879 |
| | Transformer | 1 | **1.722** | **0.679** | 1.723 | 0.709 | 1.793 | 0.699 | 2.229 | 0.759 | 2.128 | 0.705 |
| | | 15 | **1.763** | **0.680** | 2.046 | 0.727 | 2.061 | 0.719 | 2.301 | 0.763 | 2.421 | 0.781 |
| | | 30 | **1.968** | **0.735** | 2.078 | 0.753 | 2.319 | 0.772 | 2.660 | 0.826 | 2.579 | 0.814 |
| | | Avg | **1.818** | **0.698** | 1.949 | 0.730 | 2.058 | 0.730 | 2.397 | 0.783 | 2.376 | 0.767 |
| Automobile | KRR | 1 | 1.750 | 0.763 | 1.895 | 0.814 | **1.667** | **0.726** | 1.777 | 0.765 | 1.853 | 0.798 |
| | | 15 | 1.999 | 0.857 | 2.014 | 0.859 | **1.939** | **0.825** | 2.002 | 0.855 | 1.978 | 0.832 |
| | | 30 | 2.098 | 0.888 | 2.076 | 0.870 | 2.100 | 0.867 | 2.101 | 0.888 | **1.992** | **0.836** |
| | | Avg | 1.949 | 0.836 | 1.995 | 0.848 | **1.902** | **0.806** | 1.960 | 0.836 | 1.941 | 0.822 |
| | ExtraTrees | 1 | 1.692 | 0.749 | 2.141 | 0.811 | **1.616** | **0.716** | 1.770 | 0.760 | 1.944 | 0.775 |
| | | 15 | 1.990 | 0.840 | 2.239 | 0.858 | **1.956** | **0.810** | 2.036 | 0.844 | 2.274 | 0.843 |
| | | 30 | 2.173 | 0.878 | 2.315 | 0.882 | **2.169** | 0.858 | 2.181 | 0.880 | 2.352 | **0.849** |
| | | Avg | 1.952 | 0.823 | 2.232 | 0.850 | **1.914** | **0.795** | 1.996 | 0.828 | 2.190 | 0.822 |
| | MLP | 1 | 1.866 | 0.835 | 2.074 | 0.881 | **1.717** | **0.782** | 1.934 | 0.846 | 2.282 | 0.850 |
| | | 15 | 2.154 | 0.910 | 2.170 | 0.918 | 2.314 | 0.916 | **2.119** | **0.901** | 2.526 | 0.918 |
| | | 30 | 2.252 | 0.933 | 2.275 | 0.928 | 2.578 | 0.961 | **2.192** | 0.922 | 2.409 | **0.913** |
| | | Avg | 2.090 | 0.892 | 2.173 | 0.909 | 2.203 | **0.886** | **2.082** | 0.889 | 2.406 | 0.894 |
| | Transformer | 1 | 2.027 | 0.873 | 2.198 | 0.895 | 2.006 | 0.847 | 2.075 | 0.880 | **1.558** | **0.688** |
| | | 15 | 1.977 | 0.862 | 2.211 | 0.899 | 2.091 | 0.863 | 1.984 | 0.860 | **1.862** | **0.778** |
| | | 30 | 2.205 | 0.913 | 2.325 | 0.920 | 2.257 | 0.901 | **2.153** | 0.904 | 2.179 | **0.854** |
| | | Avg | 2.069 | 0.883 | 2.245 | 0.905 | 2.118 | 0.870 | 2.071 | 0.881 | **1.867** | **0.773** |
| Dysts | KRR | 1 | 0.187 | 0.300 | 0.518 | 0.556 | 0.210 | 0.332 | **0.116** | **0.250** | 0.530 | 0.575 |
| | | 15 | 0.474 | 0.528 | 0.626 | 0.623 | **0.418** | **0.490** | 0.452 | 0.524 | 0.599 | 0.611 |
| | | 30 | 0.615 | 0.614 | 0.662 | 0.645 | **0.514** | **0.553** | 0.613 | 0.623 | 0.608 | 0.617 |
| | | Avg | 0.425 | 0.481 | 0.602 | 0.608 | **0.381** | **0.458** | 0.394 | 0.465 | 0.579 | 0.601 |
| | ExtraTrees | 1 | 0.226 | 0.325 | 0.549 | 0.541 | 0.231 | **0.321** | **0.200** | 0.323 | 0.536 | 0.549 |
| | | 15 | 0.421 | 0.477 | 0.684 | 0.622 | **0.372** | **0.435** | 0.432 | 0.500 | 0.653 | 0.613 |
| | | 30 | 0.566 | 0.573 | 0.754 | 0.662 | **0.495** | **0.519** | 0.584 | 0.602 | 0.684 | 0.631 |
| | | Avg | 0.404 | 0.458 | 0.662 | 0.608 | **0.366** | **0.425** | 0.406 | 0.475 | 0.624 | 0.597 |
| | MLP | 1 | 0.191 | 0.293 | 0.499 | 0.543 | 0.195 | 0.299 | **0.143** | **0.261** | 81.408 | 1.403 |
| | | 15 | **0.466** | 0.491 | 0.622 | 0.611 | 0.483 | **0.478** | 0.556 | 0.537 | 76.533 | 1.647 |
| | | 30 | **0.640** | 0.597 | 0.664 | 0.638 | 0.677 | **0.583** | 0.818 | 0.665 | 57.319 | 1.522 |
| | | Avg | 0.432 | 0.460 | 0.595 | 0.597 | 0.452 | **0.453** | 0.506 | 0.488 | 71.753 | 1.524 |
| | Transformer | 1 | 0.432 | 0.458 | 0.713 | 0.657 | 0.391 | 0.437 | **0.370** | **0.435** | 0.521 | 0.549 |
| | | 15 | 0.550 | 0.526 | 0.724 | 0.659 | **0.462** | **0.476** | 0.533 | 0.536 | 0.586 | 0.586 |
| | | 30 | 0.666 | 0.607 | 0.740 | 0.675 | **0.569** | **0.554** | 0.717 | 0.647 | 0.640 | 0.622 |
| | | Avg | 0.549 | 0.531 | 0.726 | 0.664 | **0.474** | **0.489** | 0.540 | 0.539 | 0.582 | 0.585 |

Table 3: Full results with KSWIN. For each dataset and model family, we report MSE/MAE at prediction horizons (1, 15, 30) and their average. We compare CALIPER (ours) with fixed retraining windows of 128, 512, and 2048 steps ("Fix (·)") and with an online Incremental baseline (no explicit window retraining). The past sequence length is 30. Datasets span four heterogeneous domains (TEP, MoCap, Automobile, Dysts). Lower is better; best per column in bold (second best underlined, if applicable).

| Dataset | Model | $l_s$ | CALIPER MSE | CALIPER MAE | Fix (128) MSE | Fix (128) MAE | Fix (512) MSE | Fix (512) MAE | Fix (2048) MSE | Fix (2048) MAE | Incremental MSE | Incremental MAE |
|---|---|---|---|---|---|---|---|---|---|---|---|---|
| MoCap | KRR | 1 | **5.979** | **1.060** | 7.726 | 1.312 | 10.824 | 1.592 | 8.052 | 1.215 | 16.003 | 2.201 |
| | | 15 | **7.690** | **1.428** | 9.842 | 1.593 | 12.718 | 1.978 | 9.483 | 1.726 | 19.552 | 2.538 |
| | | 30 | **9.471** | **1.656** | 10.853 | 1.759 | 14.109 | 2.154 | 16.875 | 2.434 | 18.946 | 2.524 |
| | | Avg | **7.713** | **1.381** | 9.474 | 1.554 | 12.550 | 1.908 | 11.470 | 1.792 | 18.167 | 2.421 |
| | ExtraTrees | 1 | **7.348** | **1.374** | 13.249 | 1.867 | 10.291 | 1.750 | 9.456 | 1.599 | 35.622 | 3.337 |
| | | 15 | **9.425** | **1.597** | 19.238 | 2.274 | 13.309 | 2.018 | 10.488 | 1.781 | 44.145 | 3.777 |
| | | 30 | **9.425** | **1.670** | 17.741 | 2.276 | 14.126 | 2.060 | 11.418 | 1.869 | 33.087 | 3.298 |
| | | Avg | **8.732** | **1.547** | 16.743 | 2.139 | 12.575 | 1.942 | 10.454 | 1.750 | 37.618 | 3.471 |
| | MLP | 1 | 3.601 | 0.995 | 6.394 | 1.329 | 6.463 | 1.431 | **2.359** | **0.949** | 357.793 | 7.636 |
| | | 15 | **7.292** | **1.466** | 9.534 | 1.725 | 32.887 | 2.864 | 9.939 | 1.956 | 578.383 | 9.466 |
| | | 30 | 11.455 | **1.848** | **11.218** | 1.895 | 50.112 | 3.699 | 22.134 | 2.921 | 301.643 | 8.285 |
| | | Avg | **7.449** | **1.436** | 9.049 | 1.650 | 29.821 | 2.665 | 11.478 | 1.942 | 412.606 | 8.462 |
| | Transformer | 1 | **8.391** | **1.550** | 15.355 | 2.047 | 15.005 | 2.210 | 10.552 | 1.776 | 19.661 | 2.508 |
| | | 15 | **10.852** | **1.789** | 18.536 | 2.229 | 17.840 | 2.407 | 12.468 | 1.980 | 22.874 | 2.708 |
| | | 30 | 14.808 | **2.093** | 20.485 | 2.438 | 20.827 | 2.634 | **13.402** | 2.096 | 23.700 | 2.748 |
| | | Avg | **11.350** | **1.811** | 18.125 | 2.238 | 17.891 | 2.417 | 12.141 | 1.951 | 22.078 | 2.654 |
| TEP | KRR | 1 | 1.922 | 0.687 | 1.812 | 0.689 | 1.951 | **0.665** | 2.638 | 0.749 | **1.671** | 0.691 |
| | | 15 | 2.035 | 0.728 | 1.973 | 0.717 | 2.145 | 0.711 | 2.752 | 0.789 | **1.794** | **0.708** |
| | | 30 | 2.147 | 0.758 | 2.095 | 0.737 | 2.290 | 0.740 | 2.863 | 0.819 | **1.874** | **0.718** |
| | | Avg | 2.035 | 0.724 | 1.960 | 0.715 | 2.129 | **0.705** | 2.751 | 0.786 | **1.780** | 0.706 |
| | ExtraTrees | 1 | 1.675 | 0.637 | 1.395 | 0.636 | 1.784 | 0.654 | 2.345 | 0.735 | **1.314** | **0.621** |
| | | 15 | 1.826 | 0.686 | 1.657 | 0.680 | 2.158 | 0.716 | 2.763 | 0.801 | **1.572** | **0.663** |
| | | 30 | 1.951 | 0.721 | 1.832 | 0.701 | 2.377 | 0.751 | 3.014 | 0.842 | **1.716** | **0.678** |
| | | Avg | 1.817 | 0.681 | 1.628 | 0.672 | 2.106 | 0.707 | 2.708 | 0.792 | **1.534** | **0.654** |
| | MLP | 1 | 1.611 | 0.657 | 2.041 | 0.774 | **1.441** | **0.628** | 1.910 | 0.706 | 3.136 | 0.829 |
| | | 15 | 1.723 | 0.698 | 2.090 | 0.778 | **1.681** | **0.688** | 2.044 | 0.760 | 4.180 | 0.914 |
| | | 30 | **1.947** | **0.743** | 2.253 | 0.810 | 2.257 | 0.760 | 2.837 | 0.849 | 3.781 | 0.894 |
| | | Avg | **1.760** | 0.699 | 2.128 | 0.787 | 1.793 | **0.692** | 2.264 | 0.772 | 3.699 | 0.879 |
| | Transformer | 1 | **1.712** | **0.671** | 1.802 | 0.704 | 1.798 | 0.698 | 2.229 | 0.759 | 2.128 | 0.705 |
| | | 15 | **1.752** | **0.671** | 2.029 | 0.717 | 2.084 | 0.724 | 2.301 | 0.763 | 2.421 | 0.781 |
| | | 30 | **1.959** | **0.727** | 2.092 | 0.742 | 2.325 | 0.773 | 2.660 | 0.826 | 2.579 | 0.814 |
| | | Avg | **1.808** | **0.690** | 1.974 | 0.721 | 2.069 | 0.732 | 2.397 | 0.783 | 2.376 | 0.767 |
| Automobile | KRR | 1 | 1.778 | 0.765 | 1.864 | 0.801 | **1.678** | **0.728** | 1.778 | 0.765 | 1.853 | 0.798 |
| | | 15 | 2.002 | 0.855 | 1.982 | 0.837 | **1.934** | **0.824** | 2.002 | 0.855 | 1.978 | 0.832 |
| | | 30 | 2.100 | 0.888 | 2.026 | 0.851 | 2.099 | 0.863 | 2.100 | 0.888 | **1.992** | **0.836** |
| | | Avg | 1.960 | 0.836 | 1.957 | 0.829 | **1.904** | **0.805** | 1.960 | 0.836 | 1.941 | 0.822 |
| | ExtraTrees | 1 | 1.770 | 0.759 | 2.074 | 0.793 | **1.626** | **0.713** | 1.777 | 0.760 | 1.944 | 0.775 |
| | | 15 | 2.039 | 0.846 | 2.306 | 0.860 | **1.944** | **0.804** | 2.038 | 0.844 | 2.274 | 0.843 |
| | | 30 | 2.180 | 0.880 | 2.364 | 0.883 | **2.173** | 0.855 | 2.182 | 0.880 | 2.352 | **0.849** |
| | | Avg | 1.996 | 0.828 | 2.248 | 0.845 | **1.914** | **0.791** | 1.999 | 0.828 | 2.190 | 0.822 |
| | MLP | 1 | 1.934 | 0.845 | 2.014 | 0.862 | **1.685** | **0.771** | 1.935 | 0.846 | 2.282 | 0.850 |
| | | 15 | 2.118 | 0.901 | **2.096** | **0.896** | 2.423 | 0.914 | 2.119 | 0.901 | 2.526 | 0.918 |
| | | 30 | 2.191 | 0.922 | 2.215 | 0.916 | 2.690 | 0.962 | **2.191** | 0.922 | 2.409 | **0.913** |
| | | Avg | **2.081** | 0.889 | 2.109 | 0.891 | 2.266 | **0.882** | 2.081 | 0.889 | 2.406 | 0.894 |
| | Transformer | 1 | 2.073 | 0.879 | 2.202 | 0.880 | 1.990 | 0.837 | 2.073 | 0.879 | **1.558** | **0.688** |
| | | 15 | 1.985 | 0.860 | 2.166 | 0.877 | 2.059 | 0.858 | 1.985 | 0.860 | **1.862** | **0.778** |
| | | 30 | 2.153 | 0.905 | 2.245 | 0.896 | 2.214 | 0.893 | **2.153** | 0.905 | 2.179 | **0.854** |
| | | Avg | 2.070 | 0.881 | 2.204 | 0.884 | 2.088 | 0.863 | 2.070 | 0.881 | **1.867** | **0.773** |
| Dysts | KRR | 1 | 0.197 | 0.311 | 0.564 | 0.585 | 0.217 | 0.334 | **0.112** | **0.245** | 0.530 | 0.575 |
| | | 15 | 0.449 | 0.514 | 0.656 | 0.640 | **0.417** | **0.491** | 0.439 | 0.516 | 0.599 | 0.611 |
| | | 30 | 0.560 | 0.585 | 0.670 | 0.650 | **0.508** | **0.552** | 0.594 | 0.614 | 0.608 | 0.617 |
| | | Avg | 0.402 | 0.470 | 0.630 | 0.625 | **0.381** | 0.459 | 0.382 | **0.458** | 0.579 | 0.601 |
| | ExtraTrees | 1 | 0.227 | 0.320 | 0.587 | 0.562 | 0.217 | **0.312** | **0.196** | 0.319 | 0.536 | 0.549 |
| | | 15 | 0.397 | 0.456 | 0.723 | 0.641 | **0.351** | **0.423** | 0.423 | 0.495 | 0.653 | 0.613 |
| | | 30 | 0.520 | 0.543 | 0.757 | 0.668 | **0.474** | **0.507** | 0.570 | 0.594 | 0.684 | 0.631 |
| | | Avg | 0.381 | 0.440 | 0.689 | 0.624 | **0.347** | **0.414** | 0.396 | 0.469 | 0.624 | 0.597 |
| | MLP | 1 | 0.198 | 0.299 | 0.524 | 0.563 | 0.188 | 0.293 | **0.135** | **0.252** | 81.408 | 1.403 |
| | | 15 | 0.523 | 0.506 | 0.633 | 0.620 | **0.456** | **0.470** | 0.530 | 0.522 | 76.533 | 1.647 |
| | | 30 | 0.680 | 0.600 | 0.653 | 0.638 | **0.633** | **0.571** | 0.785 | 0.654 | 57.319 | 1.522 |
| | | Avg | 0.467 | 0.468 | 0.603 | 0.607 | **0.426** | **0.444** | 0.483 | 0.476 | 71.753 | 1.524 |
| | Transformer | 1 | 0.394 | 0.439 | 0.759 | 0.683 | 0.382 | 0.430 | **0.358** | **0.426** | 0.521 | 0.549 |
| | | 15 | 0.496 | 0.497 | 0.745 | 0.673 | **0.441** | **0.465** | 0.514 | 0.522 | 0.586 | 0.586 |
| | | 30 | 0.614 | 0.580 | 0.757 | 0.687 | **0.543** | **0.541** | 0.693 | 0.631 | 0.640 | 0.622 |
| | | Avg | 0.501 | 0.505 | 0.754 | 0.681 | **0.455** | **0.479** | 0.522 | 0.526 | 0.582 | 0.585 |

