# OpenReview forum: "When to Retrain after Drift: A Data-Only Test of Post-Drift Data Size Sufficiency"
_ICLR.cc/2026/Conference — ICLR 2026 Poster_

### Official Review · Reviewer_NZ6X · 2025-10-26

**Soundness:** 2
**Presentation:** 3
**Contribution:** 2
**Rating:** 4
**Confidence:** 3

**Summary:**

This paper addresses the problem of determining when sufficient data is available to safely retrain a model after a sudden concept drift. The authors propose CALIPER, a model-agnostic and data-only test to estimate this required post-drift data size. The core idea is grounded in the concept of "state dependence" in dynamical systems. CALIPER employs a lightweight weighted local regression (WLR) to probe the local predictability of the post-drift data window. A retraining trigger is issued when the WLR's prediction error exhibits a monotonically non-increasing trend as the locality parameter increases, conditioned on a sufficient effective sample size (ESS). The authors provide theoretical analysis linking this trigger to state dependence and learnability, and empirical results across four datasets and three model families show that CALIPER outperforms fixed-window and incremental update strategies.

**Strengths:**

1. The paper formalizes the problem of "post-drift data sufficiency," skillfully identifying the gap between drift detection and effective model adaptation. Focusing on when to retrain, rather than just if a drift occurred, is a profound and highly practical contribution to the streaming learning community.

2. The core idea of leveraging state dependence—an intrinsic data property—to infer learnability is interesting. It reframes a complex model-dependent question into a simple data-driven test. Using a lightweight WLR as a proxy is an efficient and well-justified choice for streaming environments.

3. The paper provides a theoretical foundation and well-designed experiment.

**Weaknesses:**

1. This method hinges on the assumption that the data stream is generated by a dynamical system of the form x(t+1) = f(x(t)) + noise, This assumption may not hold in many complex dynamical systems where the next state x(t+1) depends on an extended history of past states (x(t-k), ..., x(t)) [1, 2] or is affected by significant external latent factors. The paper does not provide an analysis of its robustness when this core assumption is violated, thus restricting the method's general applicability.
    [1] Learning robust spectral dynamics for temporal domain generalization, arXiv preprint arXiv:2505.12585
    [2] Continuous Temporal Domain Generalization, NeurIPS 2024.

2. While the method is model-agnostic, it implicitly assumes that data sufficiency is a property independent of the downstream model. Intuitively, however, models with varying complexity and characteristics (e.g., a linear model vs. a deep Transformer) will need different amount of samples for stable converge.  The paper does not provide an analysis of how CALIPER's estimate correlates with the true convergence points of different models, nor does it discuss how it mitigates the influence of these model-dependent data requirements.

3. The method relies on distance-based neighborhoods, which are notoriously unreliable in high-dimensional settings due to the curse of dimensionality. This could impair key components: the ESS check may require an impractically large window to meet the C(d+1) threshold, and the performance of WLR itself degrades. The experiments are conducted on relatively low-dimensional data, leaving its high-dimensional scalability unverified.

4. The post-drift window X_t is re-normalized at every time step. This dynamic scaling could itself introduce non-stationarity or mask the underlying data dynamics, especially in the presence of outliers. A more stable streaming normalization scheme might be more appropriate.

5. The ESS check is performed only at the tightest locality (θ_max), but ESS is a function of θ. This single check does not guarantee sufficient samples across the entire range of θ values tested for monotonicity, creating a potential logical inconsistency.

6. The proof sketches for Propositions 1 and 2 rely on high-level intuitive arguments (e.g., "if many pairs... a decrease could not persist") rather than formal mathematical derivations. For instance, the link between the probability of violating state dependence and the expected change in localized error needs to be made explicit.

**Questions:**

See Weaknesses.

---

> ### Author Response · Authors · 2025-11-20
>
> We thank you for the thoughtful and technically detailed review. We address your main
> concerns below.
>
> ### N1. Assumption $x(t+1)=f(x(t)) + \text{noise}$ and non-Markovian / latent factors
>
> > “This method hinges on the assumption that the data stream is generated by a dynamical system of the form x(t+1) = f(x(t)) + noise … This assumption may not hold in many complex dynamical systems where the next state x(t+1) depends on an extended history (x(t−k), ..., x(t)) or is affected by significant external latent factors.”
>
> Our theory in Section 2.3 indeed considers a one-step model
> $s(t+1) = f(s(t)) + \xi_t$ with locally Lipschitz $f$ and sub-Gaussian noise. This
> provides a clean setting to formalize state dependence and to prove that CALIPER’s
> trigger implies a learnability advantage on a suitable local region.
>
> However, CALIPER as an algorithm does not require the true system to be first-order
> in the raw observations. In practice, we apply CALIPER to feature vectors, which can
> include lagged observations or learned representations. Processes where
> $x(t+1) = g(x(t-k), \dots, x(t))$ can be represented as first-order systems by delay
> embedding the state $s(t) = [x(t-k+1), \dots, x(t)]$, and CALIPER can be applied to
> such augmented states. State dependence is then interpreted with respect to this
> representation, as in empirical dynamic modeling.
>
> We will clarify in Section 2.3 and in the limitations that:
>
> - the one-step Markovian assumption is a theoretical device for analysis,
> - in practice CALIPER can be used with arbitrary feature representations that encode
>   history, and
> - extending the formal guarantees to explicitly non-Markovian / latent-variable models
>   is an interesting direction for future work.
>
> We will also add a brief discussion relating our setting to recent work on robust
> temporal generalization.
>
> ---
>
> ### N2. Model-agnostic data sufficiency and model-dependent sample needs
>
> > “While the method is model-agnostic, it implicitly assumes that data sufficiency is a property independent of the downstream model … The paper does not provide an analysis of how CALIPER’s estimate correlates with the true convergence points of different models.”
>
> We agree that different architectures generally have different optimal sample sizes. Our
> aim is not to claim that CALIPER discovers a single universal optimum that is identical
> for all models, but rather that it detects a regime in which a broad class of reasonably
> expressive learners can retrain stably.
>
> In the current experiments, we already observe that, across four datasets and three
> model families (KRR, MLP, Transformer), the post-drift window sizes chosen by
> CALIPER are consistently close to the best-performing fixed retraining size for each
> model. In the revised version we will:
>
> - add ExtraTrees experiments, showing that the same behavior holds for a strong
>   tree-based model, and
> - include the CALIPER hyperparameter sensitivity analysis (with an MLP base
>   learner), which shows that, even when we vary $(\theta, C)$ over a wide range,
>   CALIPER’s chosen retraining times stay near the configuration-specific optimum
>   and remain competitive with fixed-window baselines.
>
> These results indicate that our state-dependence-based criterion correlates well with
> model-dependent convergence points in practice, even though the theory is stated in a
> model-agnostic way. We will clarify this interpretation in Section 3 and the discussion.
>
> ---

---

> ### Author Response · Authors · 2025-11-20
>
> ### N3. High-dimensional distance-based neighborhoods and ESS
>
> > “The method relies on distance-based neighborhoods, which are notoriously unreliable in high-dimensional settings … The ESS check may require an impractically large window, and the performance of WLR itself degrades.”
>
> We agree that naive nearest-neighbor methods can be fragile in very high dimensions.
> CALIPER mitigates this in two ways.
>
> First, we normalize distances and use the ESS gate as a conservative mechanism. At
> each time $t$, we normalize features within the current post-drift window and then
> scale pairwise distances by their mean $D$, so the scaled radii $r_i = \|x_i-x_q\|/D$
> are $O(1)$. The locality parameter $\theta$ only enters through the kernel
> $w_i(\theta) = \exp(-\theta r_i)$. For $\theta \approx 12$, points at the average
> distance already receive weight $e^{-12} \approx 6 \times 10^{-6}$, and even points at
> half the average distance receive weight below $3 \times 10^{-3}$. Thus our grid
> $\theta \in [12, 20]$ already corresponds to extremely local neighborhoods; combined
> with the ESS condition $\mathrm{ESS}(\theta_{\max}) \ge C(d+1)$, CALIPER naturally
> asks for larger windows in higher dimensions and tends to be conservative rather than
> overconfident.
>
> Second, as noted above, CALIPER operates on arbitrary feature vectors, so in
> high-dimensional raw spaces it is natural to first learn compact representations (e.g.,
> via an encoder) and apply CALIPER there. We will make this practical recommendation
> explicit and acknowledge that extreme high-dimensional settings (e.g., image streams)
> are beyond the scope of our current experiments and an interesting direction for future
> work.
>
> ---
>
> ### N4. Renormalization of the post-drift window
>
> > “The post-drift window X_t is re-normalized at every time step. This dynamic scaling could itself introduce non-stationarity or mask the underlying data dynamics, especially in the presence of outliers.”
>
> In Algorithm 1, “normalize $X_t$” refers to applying the same affine transform (per
> dimension) to all points in the current post-drift window—e.g., subtracting the window
> mean and dividing by a scale estimate. This is followed by rescaling distances by their
> mean. This step is purely for numerical stability of distances and the local regression;
> it does not mix time steps asymmetrically.
>
> CALIPER’s trigger depends only on the *relative* behavior of localized errors as
> $\theta$ varies within the same normalized window, not on absolute scale. As the
> window grows, normalization changes smoothly and affects both inputs and targets of
> WLR consistently. In practice, we found that using robust scale estimates and mild
> clipping further reduces sensitivity to outliers. We will describe this normalization
> scheme and its rationale more explicitly in Appendix G, with a short discussion in
> Section 2.2.
>
> ---
>
> ### N5. ESS check at $\theta_{\max}$ only
>
> > “The ESS check is performed only at the tightest locality (θ_max), but ESS is a function of θ … This single check does not guarantee sufficient samples across the entire range of θ values tested for monotonicity.”
>
> As discussed in the response to R-U28n, ESS$(\theta)$ is computed from the kernel
> weights $w_i(\theta)=\exp(-\theta r_i)$. As $\theta$ increases, the kernel becomes more
> local and weights concentrate on nearer points, which decreases ESS; smaller $\theta$
> makes weights more uniform and increases ESS. Therefore, $\theta_{\max}$ is the most
> local (and sparsest) neighborhood on the grid: if $\mathrm{ESS}(\theta_{\max}) \ge C(d+1)$,
> then ESS$(\theta)$ is even larger for all smaller $\theta$.
>
> Thus, checking ESS only at $\theta_{\max}$ is a worst-case test that guarantees sufficient
> effective samples for all localities used in the monotonicity check. We will make this
> monotonicity argument explicit in Section 2.2 to remove the apparent inconsistency.
>
> ---
>
> ### N6. Proof sketches and rigor
>
> > “The proof sketches for Propositions 1 and 2 rely on high-level intuitive arguments … The link between the probability of violating state dependence and the expected change in localized error needs to be made explicit.”
>
> We appreciate this comment. Our intention in the main text was to provide only
> high-level proof sketches for Propositions 1 and 2 in order to keep the presentation
> readable, while giving the full mathematical details in the appendix.
>
> In particular, the appendix already contains complete proofs of both propositions,
> including the explicit link between violations of state dependence and changes in
> localized prediction error, as well as the corresponding concentration arguments.
> All assumptions and intermediate steps are stated there in full.
>
> Given the page limits and the risk of overloading the main body with technical
> details, we prefer to keep the current level of detail unchanged and to refer
> interested readers to the appendix for the full derivations. We hope this resolves the
> concern about rigor while preserving the accessibility of the exposition.

---

> > ### Comment · Reviewer_NZ6X · 2025-11-23
> >
> > Thank you for the responses. The clarifications regarding the Markovian assumption, ESS monotonicity, high-dimensional behavior, and the expanded analysis address most of my concerns. With these additions, I believe the contribution is now substantially strengthened, and I am raising my score to 6.

---

> > > ### Author Response · Authors · 2025-11-23
> > >
> > > We sincerely appreciate your follow-up and are glad that our clarifications on the Markovian assumption, ESS monotonicity, high-dimensional behavior, and the expanded analysis have addressed your concerns. Thank you again for your thoughtful review and for taking the time to reconsider and update your evaluation.

---

### Official Review · Reviewer_U28n · 2025-10-27

**Soundness:** 3
**Presentation:** 2
**Contribution:** 2
**Rating:** 2
**Confidence:** 3

**Summary:**

The paper proposes a method for determining the right time to retrain/adapt a model after concept drift has occurred. The proposed method is computational efficient because it only uses the data from the data stream together with some hyperparameters.

**Strengths:**

Strengths
- Clear contribution with potentially high impact, well-grounded in the literature
- Formal analysis of the proposed algorithm
- Overall, the paper is well structured. Mostly easy to read, with some exceptions (see "Weaknesses" for suggestions on how to improve it)

**Weaknesses:**

The state dependence looks like a pretty strong assumption, as it assumes continuity of the state transition function. I am not sure if this is safe to assume. In particular, in high-dimensional settings. This is also reflected in Section 2.3. I miss a critical discussion on those assumptions, apart from Appendix C. Maybe one could also run experiments on datasets that differ in their dimensionality to get a better feeling for how the method performs in such cases.

Problem 1: I do not understand why minimising the generalisation loss links to the data-side stopping criterion R. If I understand it correctly, they are supposed to model the same thing -- however, the generalisation error is difficult to compute in an online approach, right? If so, it is just a change of notation. I suggest clarifying this to avoid any potential confusion on the reader's side.

Line 216: If I understood it correctly, the following steps have to be executed for every theta in the locality grid, right? If so, please state this explicitly!

What is the reasoning behind line 223 "Proceed only if ..."? Do you want to ensure that the data set contains enough samples (see statement in line 238)? I suggest adding a short clarification. In general, it is important to ensure that all steps in the algorithm, the reader understands why certain things are done.

A few comments on the GitHub:
- Please provide more information in the README, in particular, describing the outline of the repository (i.e., where to find what).
- Please put more comments and docstrings in the code

Minor:
- Acronym WLR in Figure 1.c not introduced -- becomes obvious later in the paper. However, I recommend introducing it when it is first used.
- Problem 1: While I am familiar with epsilon-delta notions, it might be good to add a brief explanation to make the notation more accessible.
- I suggest moving the proof sketches into the appendix -- they kind of interrupt the reading flow

**Questions:**

See weaknesses

---

> ### Author Response · Authors · 2025-11-20
>
> We appreciate your detailed comments and suggestions regarding assumptions, clarity
> of the formalization, and practical aspects.
>
> ### U1. Strength of state-dependence assumptions and high-dimensional settings
>
> > “The state dependence looks like a pretty strong assumption, as it assumes continuity of the state transition function. I am not sure if this is safe to assume, in particular in high-dimensional settings. I miss a critical discussion on those assumptions, apart from Appendix C.”
>
> Our theoretical analysis adopts a stylized but standard setting: a $d$-dimensional
> time series $\{s(t)\}$ generated by
> $$
> s(t+1) = f(s(t)) + \xi_t,
> $$
> with locally Lipschitz $f$ and sub-Gaussian noise. These assumptions are used only to
> analyze CALIPER’s trigger—i.e., to formally connect the monotone locality-curve plus
> ESS condition to state dependence and learnability—not to define the algorithm itself
> or its domain of application.
>
> In practice, the observed state $x(t)$ can be taken to include a short history of
> the raw stream (e.g., via delay embedding), so a first-order Markov model can hold for
> this augmented state even when the original process depends on
> $(x(t-k), \dots, x(t))$. Moreover, CALIPER is applied to arbitrary feature vectors,
> which may be raw observations, delay-embedded states, or learned representations from
> an upstream encoder.
>
> We will move part of the discussion currently in the appendix into Section 2.3 and add a
> paragraph explicitly stating that:
>
> - these conditions are regularity assumptions for the analysis, not hard requirements
>   for applying the algorithm, and
> - in high-dimensional settings, it is natural to operate on suitable representations and
>   let the ESS gate act as a conservative safeguard when local neighborhoods are too
>   sparse.
>
> ---
>
> ### U2. Relation between Problem 1 (generalization loss) and $R(X_t)$
>
> > “I do not understand why minimising the generalisation loss links to the data-side stopping criterion R. If I understand it correctly, they are supposed to model the same thing -- however, the generalisation error is difficult to compute in an online approach, right? If so, it is just a change of notation.”
>
> We thank you for pointing out this potential confusion. Our intention is as follows:
>
> - Problem 1 defines an ideal stopping time in terms of the unobservable
>   generalization loss $L_{\text{gen}}$ of a downstream learner. This quantity cannot be
>   computed in streaming.
> - CALIPER then constructs a computable data-side surrogate $R(X_t)$ based
>   on the ESS gate and the monotonicity of localized one-step prediction error under
>   weighted local regression.
>
> Thus, $R(X_t)$ is not merely a reparameterization of $L_{\text{gen}}$, but a provably
> aligned proxy under our assumptions: Propositions 1–2 show that when $R(X_t)=1$,
> the post-drift window lies in a state-dependent regime where data-dependent
> generalization bounds become tighter, connecting the surrogate to improved retraining
> performance.
>
> We will revise Sections 2.1–2.2 to explicitly emphasize this “ideal vs surrogate”
> distinction and avoid any impression that we are simply changing notation.
>
> ---
>
> ### U3. Algorithmic details: per-$\theta$ steps and “Proceed only if …”
>
> > “If I understood it correctly, the following steps have to be executed for every theta in the locality grid, right?”
> > “What is the reasoning behind line 223 ‘Proceed only if ESS(θ_max) ≥ C(d+1)’?”
>
> You are correct that, at each time $t \ge t_s+1$, CALIPER executes the weighted local
> regression and error computation for all $\theta \in \Theta$ on the current
> post-drift window, updating locality-indexed statistics $E(t,\theta)$. We will state this
> explicitly in the algorithm description.
>
> The condition “Proceed only if $\mathrm{ESS}(\theta_{\max}) \ge C(d+1)$” acts as a
> safety gate. ESS$(\theta)$ is defined from the kernel weights
> $w_i(\theta) = \exp(-\theta r_i)$. As $\theta$ increases, the kernel becomes more local
> and weights concentrate on fewer neighbors, which decreases ESS; conversely,
> smaller $\theta$ yields more uniform weights and larger ESS. Therefore, checking ESS
> only at $\theta_{\max}$ corresponds to a worst-case test: if
> $\mathrm{ESS}(\theta_{\max}) \ge C(d+1)$, then $\mathrm{ESS}(\theta) \ge C(d+1)$ for all
> $\theta \in \Theta$. This guarantees sufficient effective samples for all localities used in
> the monotonicity test. We will add this monotonicity argument directly after the ESS
> definition.
>
> ---

---

> > ### Author Response · Authors · 2025-11-20
> >
> > ### U4. Code and repository documentation
> >
> > > “Please provide more information in the README … Please put more comments and docstrings in the code.”
> >
> > We appreciate these practical suggestions. For the camera-ready version, we will:
> >
> > - expand the README to describe the repository structure (where to find CALIPER’s
> >   implementation, experiment scripts, and data preprocessing),
> > - add usage examples for reproducing each figure and table, and
> > - include additional comments and docstrings in the main modules.
> >
> > We will mention these improvements in the reproducibility statement.
> >
> > ---
> >
> > ### U5. Minor comments
> >
> > - **WLR acronym in Figure 1(c).** We will introduce “weighted local regression
> >   (WLR)” the first time it appears in the text, before using the acronym in figures.
> > - **$\varepsilon$–$\delta$ notation in Problem 1.** We will add a brief explanatory
> >   sentence stating that $\varepsilon > 0$ is a target error level and $\delta \in (0,1)$ is the
> >   tolerated failure probability, to make the notation more accessible.
> > - **Proof sketches.** We will move the detailed proof sketches from the main text into
> >   the appendix and retain only short high-level summaries in Section 2.3, improving
> >   the reading flow while keeping the mathematical details available.

---

> > > ### Comment · Reviewer_U28n · 2025-11-21
> > >
> > > Thanks for your rebuttal and the revised manuscript. I read both, and they indeed clarify my questions as well as most of my concerns. Therefore, I will increase my score.
> > >
> > > The potential issue of high-dimensional domains remains my only concern. I agree with the authors that using a "reasonable" representation might help, but it would be good to see at least one example of such a case in the experiments. This is why I can not further increase my score. If this paper is not accepted, I recommend adding some experiments on higher-dimensional domains to it before resubmitting it -- I believe that this would increase the acceptance chances.

---

> > > > ### Author Response · Authors · 2025-11-21
> > > >
> > > > We sincerely appreciate your recognition and are pleased to hear that our responses have addressed your concerns and contributed to the increased score. We thank you again for your kind suggestions and dedication to the review process.

---

### Official Review · Reviewer_JXVC · 2025-11-01

**Soundness:** 3
**Presentation:** 3
**Contribution:** 4
**Rating:** 6
**Confidence:** 1

**Summary:**

This paper focuses on handling the sudden drift in streaming data and tries to explore when to retrain after drift. A method called CALIPER has been developed for detecting concept drift occurrence and stable retraining. And a theoretical analysis of the proposed method has been given for fundamental support. The experiment on several datasets and benchmarks has been conducted, and the experiment results show the performance of the proposed method.

**Strengths:**

1. The idea of "when to retrain after drift" is interesting, the previous works usually retrain directly when drift is detected. This work focuses on identifying the right time to retrain the model to help enhance the learning stability.

2. The proposed method is well designed with a detailed theoretical analysis, and the experiment is sufficient and reflects the learning performance of the proposed method.

**Weaknesses:**

1. In the experiment, models like MLP and transformer have been chosen for comparison. I think tree-based models should also be chosen for comparison, since they are commonly used in traditional concept drift learning.

2. The author only compares the proposed method with ADWIN, which is a traditional drift detection method, more comparisons with recently proposed drift detection methods are required.

3. A parameter analysis is needed to show the robustness of the proposed method.

**Questions:**

1. The author should clarify the difference between the proposed method with traditional concept drift learning methods to enhance the novelty.

2. The experiment should not only focus on MLP and Transformer, but also should focus on tree-based model, which performs excellently in concept drift learning.

3. More comparison on the recently proposed concept drift detection method is required.

---

> ### Author Response · Authors · 2025-11-20
>
> We thank you for the positive assessment of our contribution and for the
> constructive suggestions on experiments and robustness.
>
> ### Q1. Difference to traditional concept drift learning / novelty
>
> > “The author should clarify the difference between the proposed method with traditional concept drift learning methods to enhance the novelty.”
>
> Classical concept drift methods such as ADWIN or KSWIN focus on
> **drift detection**: they monitor the stream to decide *if* and *when* a
> distributional change has occurred. Once a drift is detected, standard practice is to either
>
> - retrain immediately on all available post-drift data, or
> - continue updating with a fixed window size or learning rate.
>
> In contrast, our contribution is explicitly **post-drift data sufficiency**. In Section 2.1
> we formalize an ideal stopping rule based on the generalization loss of a
> downstream learner, defining the earliest time when retraining would be safe. Since
> this ideal rule is not computable in streaming, CALIPER introduces a data-only
> surrogate $R(X_t)$ that fires when the post-drift window exhibits sufficient state
> dependence and ESS. CALIPER can be plugged into any detector–learner pipeline and
> answers a question that existing detectors leave open: given a drift alarm, when has
> enough post-drift data accumulated to justify retraining?
>
> We will add a short paragraph in Section 1 clarifying this distinction and explicitly
> positioning CALIPER as complementary to ADWIN/KSWIN and related detectors.
>
> ---
>
> ### Q2. Tree-based models
>
> > “The experiment should not only focus on MLP and Transformer, but also tree-based model, which performs excellently in concept drift learning.”
>
> We agree that tree-based learners are widely used in concept drift settings. In the
> revised manuscript, we will include additional experiments using an Extremely
> Randomized Trees (ExtraTrees) regressor as a strong tree-based base learner.
>
> We keep the detectors (ADWIN and KSWIN), datasets, and CALIPER hyperparameters
> identical to the main experiments, sweep several fixed post-drift window sizes, and
> compare them to CALIPER. Across all datasets and detectors, CALIPER’s estimated
> window sizes again remain close to the empirically optimal fixed choice for ExtraTrees,
> and its post-drift error matches or improves upon the best fixed window. These results
> confirm that CALIPER behaves similarly for tree-based models as for KRR, MLP, and
> Transformers. We will summarize these findings in the appendix and refer to them in
> Section 3.
>
> ---
>
> ### Q3. More comparisons with drift detection methods
>
> > “The author only compares the proposed method with ADWIN, which is a traditional drift detection method, more comparisons with recently proposed drift detection methods are required.”
>
> Our experiments already use two detectors: ADWIN and the more recent KSWIN.
> CALIPER itself is detector-agnostic: it only consumes the post-drift window produced
> by any detector and does not depend on how the drift was detected. As such, any
> detection method—older or more recent—could be plugged into the same pipeline, and
> CALIPER’s role (estimating when the post-drift window is sufficient) would remain
> unchanged.
>
> Given space constraints, our goal was not to benchmark detectors, but to show that
> for a given detector, CALIPER improves the choice of retraining time relative to fixed
> windows or incremental updates. We will clarify this design choice in the text and
> briefly discuss how other detectors could be integrated in practice.
>
> ---

---

> ### Author Response · Authors · 2025-11-20
>
> ### Parameter analysis / robustness
>
> > “A parameter analysis is needed to show the robustness of the proposed method.”
>
> We address robustness in two ways:
>
> 1. **Shared CALIPER hyperparameters across all experiments.**
>    We already use a single locality grid $\Theta$ and ESS threshold $C(d+1)$ across all
>    datasets and base learners, without per-dataset tuning. Despite this, CALIPER
>    matches or surpasses the best fixed retraining size on all benchmarks. We will make
>    this “single shared configuration” explicit in Section 3 and Appendix G.
>
> 2. **Sensitivity of CALIPER’s own hyperparameters (with an MLP base learner).**
>    We further conduct a hyperparameter sensitivity analysis of CALIPER itself when
>    using an MLP as the base learner. For each dataset and each detector (ADWIN and
>    KSWIN), we vary the ESS threshold $C \in \{2,3,4\}$ and the maximum locality
>    parameter $\theta_{\max} \in \{12.0, 14.0, 16.0, 18.0, 20.0\}$, and record the resulting
>    post-drift mean absolute error (MAE). An appendix figure (MLP base learner)
>    plots MAE as a function of $C$ and $\theta_{\max}$; across all four datasets and both
>    detectors, the curves are nearly flat, indicating that CALIPER’s selected window
>    sizes and post-drift errors are stable over this wide range of settings. This empirical
>    behavior is consistent with our design choices for the locality grid and ESS gate.
>
>    More concretely, in all experiments we normalize pairwise distances by the mean
>    distance $D$, so that scaled radii $r_i = \|x_i - x_q\| / D$ are typically $O(1)$ and
>    $\theta$ only enters through the kernel $w_i(\theta) = \exp(-\theta r_i)$. For
>    $\theta \approx 12$, points at the average distance already receive weight
>    $e^{-12} \approx 6 \times 10^{-6}$, and even points at half the average distance
>    have weight below $3 \times 10^{-3}$. Thus our grid $\theta \in [12, 20]$ already
>    corresponds to very tight neighborhoods that effectively restrict the regression to a
>    small ball around the query point $x_q$, while still retaining enough effective
>    samples to satisfy $\mathrm{ESS}(\theta_{\max}) \ge C(d+1)$. Exploring much larger
>    values of $\theta$ would push CALIPER into an extremely small-neighborhood
>    regime where ESS becomes unstable and the theoretical guarantees—which require a
>    sufficiently populated local ball—no longer provide additional benefits. This
>    explains why scanning up to $\theta_{\max} \in [12, 20]$ is consistent with the
>    theoretical design and empirically sufficient in our sensitivity study.
>
>    *On the choice of $C \in [2,4]$.* In CALIPER, the ESS gate
>    $\mathrm{ESS}(\theta_{\max}) \ge C(d+1)$ enforces that the local WLR has at least $C$
>    effective samples per parameter. Standard finite-sample analyses of local and linear
>    regression yield a variance term of the form $\sigma^2 (d+1)/\mathrm{ESS}$, so imposing
>    $\mathrm{ESS} \ge C(d+1)$ effectively bounds this term by $\sigma^2 / C$. Thus moving
>    from $C=1$ (just-identifiable) to $C=2$ or $C=4$ already reduces this variance
>    contribution by a factor of 2–4, which we found sufficient for stable retraining.
>    Importantly, increasing $C$ further yields only diminishing returns in terms of this
>    post-drift variance (e.g., from $C=4$ to $C=8$) but linearly increases the number
>    of post-drift samples required before retraining. In a streaming setting this directly
>    translates into a longer period during which a stale pre-drift model is used,
>    increasing the cumulative loss on the stream. We therefore restrict $C$ to the range
>    2–4, which is both theoretically motivated (sufficient effective samples per parameter)
>    and empirically robust across datasets.

---

> > ### Comment · Reviewer_JXVC · 2025-11-27
> >
> > I have read the response carefully, the authors have addressed all my questions, I have no other concerns. And I will raise the score. Thank you!

---

> > > ### Author Response · Authors · 2025-11-27
> > >
> > > We sincerely appreciate your follow-up and are pleased to hear that our responses and the revised manuscript have addressed all of your questions. Thank you again for your careful reading, constructive feedback, and for kindly updating your evaluation.

---

### Author Response · Authors · 2025-11-20
**Global response**

We thank all reviewers for their careful reading and constructive feedback. We are
encouraged that all three reviewers find the problem of *when to retrain after
concept drift* important and the proposed idea interesting:

- R-JXVC: “The idea of ‘when to retrain after drift’ is interesting … The proposed
  method is well designed with detailed theoretical analysis, and the experiment is
  sufficient.”
- R-U28n: “Clear contribution with potentially high impact, well-grounded in the
  literature … formal analysis of the proposed algorithm … overall, the paper is well
  structured.”
- R-NZ6X: “Focusing on when to retrain, rather than just if a drift occurred, is a
  profound and highly practical contribution … The core idea … is interesting …
  The paper provides a theoretical foundation and well-designed experiment.”

Our work targets a problem that is different but complementary to classical *drift
detection*. Existing detectors (e.g., ADWIN, KSWIN) decide *whether and when*
a drift has occurred. In streaming settings, however, practitioners face a further,
unresolved trade-off *after* a drift alarm:

- waiting for more post-drift samples typically allows a sufficiently expressive learner
  to achieve lower generalization error after retraining, but
- waiting too long means continuing to use a stale pre-drift model, accumulating
  prediction error on the stream.

The central question we address is therefore:

**Given that a drift alarm has fired, what is the earliest time at which the
post-drift window contains enough information to safely retrain?**

Rather than trying to identify the exact model-specific sample size, we seek a
*model-agnostic, data-side criterion* that detects when the post-drift data have
entered a locally learnable regime of the underlying system. CALIPER provides such
a stopping rule by testing for state dependence under an effective sample size (ESS)
gate, using only the data stream and remaining agnostic to the downstream
architecture and the choice of drift detector.

In the revised version, we will (i) clarify this problem setting and the role of our
assumptions, (ii) add experiments with a tree-based learner and a sensitivity analysis of
CALIPER’s own hyperparameters, and (iii) improve algorithmic clarity and code
documentation. These changes directly address the reviewers’ main concerns while
keeping the core method unchanged.

Again, we thank all reviewers for their insightful comments. We believe that the
clarifications, additional experiments (tree-based learner and hyperparameter
sensitivity), and small structural adjustments outlined above will significantly improve
the paper.

---

### Author Response · Authors · 2025-11-28
**Summary for the Newly Assigned Area Chair**

We would like to briefly summarize the current status of the reviews and how our responses address the main concerns.

All three reviewers view the problem and our contribution positively:

- **R-JXVC** finds the idea of “when to retrain after drift” interesting and notes that the method is well designed with detailed theoretical analysis and sufficient experiments.
- **R-U28n** highlights a clear contribution with potentially high impact, well grounded in the literature, with a formal analysis of the proposed algorithm and an overall well-structured presentation.
- **R-NZ6X** emphasizes that focusing on *when* to retrain after a detected drift, rather than only deciding whether a drift occurred, is a practical and important contribution, and finds the core idea of using state dependence with a lightweight WLR test interesting.

In our rebuttal and revised manuscript, we have:

- **Clarified the problem setting and assumptions**, including the Markov/representation assumptions and the relation between the ideal stopping rule based on generalization loss and our data-side surrogate criterion. We have also moved parts of the theoretical discussion from the appendix into the main text for better accessibility.
- **Added experiments and analyses**, including an ExtraTrees-based tree learner, a sensitivity analysis of CALIPER’s own hyperparameters, and a more detailed discussion of high-dimensional behavior and the role of the ESS gate.
- **Improved algorithmic clarity and code documentation**, by more explicitly describing the per-θ steps and the rationale behind the ESS condition, and by committing to a clearer repository structure and additional comments/docstrings in the implementation.

In their follow-up comments, **all three reviewers** stated that **our responses had resolved their main questions and substantially strengthened the paper**, and that this led them to reconsider the work and **raise their scores**. Concretely, before the rollback, this was reflected in their updated ratings: R-JXVC from 6 to 8, R-U28n increased their score from 2 to 6, and R-NZ6X from 4 to 6. Thus, prior to the rollback, all three reviewers’ post-discussion assessments placed the paper at least on the positive side of the borderline. Regarding higher-dimensional settings, the reviewers and we share the view that applying CALIPER to suitable, “reasonable” representations is a natural and practical way to mitigate potential issues, and explicit experiments in such high-dimensional, representation-based settings were highlighted as a promising direction for future work that could further enhance the contribution.

---

### Meta-Review · Area_Chair_u8DW · 2026-01-06

**Summary:**

One important issue raised by rev. U28n and NZ6X is the Markovian assumption on the state of the model. Also, the same revs. asked whether the model would work on high-dimensional data.
In addition, rev. JXVC  asked for a sensitivity analysis of the hyperparameters of the proposed model and an evaluation on a tree learner.
Finally, many clarifications and improvements on the original presentation of the manuscript were requested by the three reviewers.

**Reviewer Concerns:**

The concern about the Markovian assumption was solved by explaining that even if a time series is not Markovian, the model could use memory and consider multiple time steps as a state, resorting to a Markovian system.
The issue about the behaviour of the proposed approach with high-dimensional data was not fully solved. The authors assumed that with a "reasonable" representation, the proposed approach should still work; however, I agree with rev. U28n that an experiment on high-dimensional data would have fully clarified the issue.
Concerns about details and presentation were solved in the rebuttal.
Overall, I agree with the reviewers that the paper deserves publication, but for the final version of the paper, I would also like to see some experiments on high-dimensional data.

**Reviewer Scores:**

- Rev JXVC: 6 -> 8
Authors answered well to all rev. questions by adding experiments on a tree-based model, explaining the difference with previous drift-detection approaches and showing the sensitivity of the model's hyperparameters.
- Rev. U28n: 2 -> 6
Also, in this case, the authors did a good job in answering all rev. questions. The only one that remains is the behaviour of the method for high-dimensional data. This is why rev. did not go higher than 6.
- Rev NZ6X: 4 -> 6
Also in this case authors' answers were convincing and rev. increased their score to 6.

---

### Decision · Program_Chairs · 2026-01-26

Accept (Poster)